# Exploration-Exploitation in Multi-Agent Competition: Convergence with Bounded Rationality

**Stefanos Leonardos, Georgios Piliouras**
Singapore University of Technology and Design
`{stefanos_leonardos;georgios}@sutd.edu.sg`

**Kelly Spendlove**
University of Oxford
`spendlove@maths.ox.ac.uk`

## Abstract

The interplay between exploration and exploitation in competitive multi-agent learning is still far from being well understood. Motivated by this, we study smooth Q-learning, a prototypical learning model that explicitly captures the balance between game rewards and exploration costs. We show that Q-learning always converges to the unique quantal-response equilibrium (QRE), the standard solution concept for games under bounded rationality, in weighted zero-sum polymatrix games with heterogeneous learning agents using positive exploration rates. Complementing recent results about convergence in weighted potential games [16, 34], we show that fast convergence of Q-learning in competitive settings obtains regardless of the number of agents and without any need for parameter fine-tuning. As showcased by our experiments in network zero-sum games, these theoretical results provide the necessary guarantees for an algorithmic approach to the currently open problem of equilibrium selection in competitive multi-agent settings.

## 1 Introduction

Zero-sum games and variants thereof are arguably amongst the most well studied settings in game theory. Indeed much attention has focused on the class of strictly competitive games [4], i.e., two player games such that when both players change their mixed strategies, then either there is no change in the expected payoffs, or one of the two expected payoffs increases and the other decreases.[1] According to Aumann [4], *"Strictly competitive games constitute one of the few areas in game theory, and indeed in the social sciences, where a fairly sharp, unique prediction is made."* The unique prediction, of course, refers to the min-max solution and the resulting values guaranteed to both agents due to the classic work of von Neumann [53].

Unfortunately, when we move away from the safe haven of two-agent strictly competitive games, a lot of these regularities disappear. For example, in multi-agent variants of zero-sum and strictly competitive games, several critical aspects of the min-max theorem collapse [12]. Critically, *Nash Equilibrium* (NE) payoffs need not be unique. In fact, there can be continua of equilibria with the payoff range of different agents corresponding to positive measure sets. Furthermore, NE strategies need not be exchangeable (i.e., mixing-matching strategies from different Nash profiles does not lead to a Nash) nor max-min. Thus, network competition is not only significantly harder, but poses qualitatively different questions than two-agent competition.

Nevertheless, and despite the intense recent interest inspired by Artificial Intelligence (AI) and Machine Learning (ML) applications such as generative adversarial networks and actor-critic systems to understand learning dynamics in zero-sum games and even network variants thereof [19, 22, 27], so far, there has been no systematic study of how agents should deal with uncertainty of the resulting

---

[1]In fact, as recent work has established these strictly competitive games are formally equivalent to weighted zero-sum games, i.e., affine transformations of zero-sum games [1].

35th Conference on Neural Information Processing Systems (NeurIPS 2021).

payoffs in such games. In these settings, the use of purely optimization-driven, regret-minimizing algorithms is no longer equally attractive as in the two-player case. The multiplicity of equilibria and the lack of unique value give rise to a non-trivial problem of *equilibrium selection* and learning agents face the fundamental dilemma between exploration and exploitation [15, 41, 11, 59]. These considerations drive our motivating question:

*Are there exploration-exploitation dynamics that provably converge in networks of strictly competitive games? How do they behave in settings with multiple, payoff diverse Nash equilibria?*

**Model and Results.** We study a well-known smooth variant of Q-learning [55, 54], with softmax or Boltzmann exploration, one of the most fundamental models of exploration-exploitation in multi-agent systems (MAS), termed Boltzmann or *smooth Q-learning* [51, 48]. Informally (see Section 3 for the rigorous definition), each agent $k$ updates their choice distribution $x = (x_i)$ according to the rule $\dot{x}_i/x_i = (r_i - \bar{r}) - T_k(\ln x_i - \sum_j x_j \ln x_j)$, where $r_i, \bar{r}$ denote agent $k$'s rewards from action $i$ and average rewards, respectively, given all other agents' actions and $T_k$ is agent $k$'s exploration rate.

In our main result, we show convergence of Q-learning to *Quantal Response Equilibria* (QRE), the prototypical extension of Nash equilibria for games with bounded rationality [36], in multi-agent/network generalizations of strictly competitive games [13, 12]. As long as all exploration rates are positive, we prove via a global Lyapunov argument that the Q-learning dynamic converges pointwise to a unique QRE regardless of initial conditions and regardless of the number of the Nash equilibria of the original network game (Theorem 4.1). Related to the above, we demonstrate how exploration by all agents leads to equilibrium selection. Thus, this long-standing open problem ([32, 45, 43]) becomes tractable in practice due to the theoretical guarantees of fast convergence to QRE that we provide in Theorem 4.1 for this class of competitive games. In fact, Theorem 4.1 is in some sense tight, as there exist network competitive settings whose dynamics lead to limit cycles if not all of the agents are performing exploration (see experiments and discussion in Section 5).

**Other Related Works.** The variant of Q-learning that we study has recently received a lot of attention due to its connection to evolutionary game theory [47, 31, 57]. It has been also extensively studied in the economics and reinforcement learning literature under various names, see e.g., [2, 46] and [28, 39]. Recent works demonstrate that it is possible to show convergence of the Q-learning dynamics in multi-agent *cooperative* settings and to select highly desirable equilibria via complex bifurcation phenomena [34].

On the other hand, competitive multi-agent systems constitute one of the current frontiers in AI and ML research. Many recent works investigate the complex behavior of competitive game theoretic settings [44, 9, 8, 7, 37], focusing on carefully designed convergent algorithms, e.g., *optimism* [19, 20, 56], extra-gradient methods [40, 3, 26], regularization [43], momentum adapted dynamics [23], or symplectic integration schemes [6]. However, despite the theoretical progress and the impressive results in the empirical front [49, 52, 33], the literature on equilibrium selection seems to have received little attention so far. Prior works have focused on equilibrium selection in cooperative AI or have theoretically studied competition in 2-agent zero-sum games in which equilibrium selection is irrelevant [17, 18]. To our knowledge, we are the first paper to explicitly study both theoretically and experimentally multi-agent competitive settings without uniquely defined values.

## 2 Game-Theoretic Model

A *polymatrix or separable network game*, $\Gamma = \left( (V, E), (S_k, w_k)_{k \in V}, (\mathbf{A}_{kl})_{[k,l] \in E} \right)$, consists of a graph $(V, E)$, where $V = \{1, 2, \ldots, n\}$ is the set of players (or agents), and $E$ is a set of pairs, $[k, l]$, of players $k \neq l \in V$. Each player, $k \in V$, has a finite set of actions (or strategies) $S_k$ with generic elements $s_k \in S_k$ (depending on the context, sometimes we will also write $i$ or $j \in S_k$). Players may also use mixed strategies (or choice distributions) $\mathbf{x}_k = (x_{ki})_{i \in S_k} \in \Delta_k$, where $\Delta_k$ is the simplex in $\mathbb{R}^{|S_k|}$, i.e., $\sum_{i \in S_k} x_{ki} = 1$, and $x_{ki} \geq 0$, for any $\mathbf{x}_k \in \Delta_k$. The *interior* of $\Delta_k$ is the set of all points $\mathbf{x}_k \in \Delta_k$ with $x_{ki} \in (0, 1)$ for all $i \in S_k$. All points of $\Delta_k$ that are not in the interior, are called *boundary points*. We will write $\mathbf{x} = (\mathbf{x}_1, \mathbf{x}_2, \ldots, \mathbf{x}_n)$ or $\mathbf{x} = (\mathbf{x}_k, \mathbf{x}_{-k})$ for a mixed strategy profile $\mathbf{x} \in \Delta := \prod_{k \in V} \Delta_k$, where $\mathbf{x}_{-k} \in \Delta_{-k} = \prod_{l \neq k \in V} \Delta_l$ is the vector of mixed strategies of all players $l \in V$ other than $k$.

Each edge $[k, l] \in E$ defines a two-player game with payoff matrices $\mathbf{A}_{kl} \in \mathbb{R}^{|S_k| \times |S_l|}$ and $\mathbf{A}_{lk} \in \mathbb{R}^{|S_l| \times |S_k|}$. The elements $a_{kl}(s_k, s_l)$ of a matrix $\mathbf{A}_{kl}$ denote the payoffs of player $k$ when players $k$ and $l$ use pure actions $s_k \in S_k$ and $s_l \in S_l$, respectively. Each player $k \in V$ chooses a strategy (mixed or pure) and plays that strategy in *all* games $[k, l] \in E$. Thus, the payoff of player $k$ at the pure strategy profile $\mathbf{s} = (s_1, \ldots, s_n) \in S := \prod_{k \in V} S_k$ is $u_k(\mathbf{s}) = \sum_{[k,l] \in E} a_{kl}(s_k, s_l)$. Similarly, the expected reward of player $k$ in the mixed strategy profile $\mathbf{x} \in \Delta$ is

$$u_k(\mathbf{x}) := \sum_{[k,l] \in E} \mathbf{x}_k^\top \mathbf{A}_{kl} \mathbf{x}_l = \mathbf{x}_k^\top \left( \sum_{[k,l] \in E} \mathbf{A}_{kl} \mathbf{x}_l \right). \tag{1}$$

It will be convenient to write $r_{ki}(\mathbf{x}_{-k}) := u_k(i, \mathbf{x}_{-k}) = \sum_{[k,l] \in E} \{\mathbf{A}_{kl} \mathbf{x}_l\}_i$ (where $\{v\}_i$ to denotes the $i$-th element of a vector $v$), for the reward of pure action $i \in S_k$ of player $k$ when all other players use the strategy profile $\mathbf{x}_{-k}$ and $r_k(\mathbf{x}_{-k}) := (r_{ki}(\mathbf{x}_{-k}))_{i \in S_k}$ for the resulting reward vector of all pure actions of agent $k$, respectively. Using this notation, the expected reward of player $k \in V$ at the mixed strategy profile $\mathbf{x} = (\mathbf{x}_k, \mathbf{x}_{-k})$ can be compactly expressed as $u_k(\mathbf{x}) = \mathbf{x}_k^\top r_k(\mathbf{x}_{-k})$.

**Weighted Zero-Sum Polymatrix Games.** $\Gamma$ is called a *weighted or rescaled zero-sum polymatrix game* [12], if there exist positive constants $w_1, w_2, \ldots, w_n > 0$, so that

$$\sum_{k \in V} w_k u_k(\mathbf{x}) = 0, \quad \text{for all } \mathbf{x} \in \Delta. \tag{2}$$

By summing over the edges in $E$ (instead of the players in $V$), we can equivalently express the weighted zero-sum property as

$$\sum_{[k,l] \in E} \left[ w_k \mathbf{x}_k^\top \mathbf{A}_{kl} \mathbf{x}_l + w_l \mathbf{x}_l^\top \mathbf{A}_{lk} \mathbf{x}_k \right] = 0, \quad \text{for all } \mathbf{x} \in \Delta. \tag{3}$$

**Nash Equilibrium.** A strategy profile (tuple of mixed strategies), $\mathbf{p} = (p_k)_{k \in V} \in \Delta$, with $\mathbf{p}_k = (p_{ki})_{i \in S_k} \in \Delta_k$ for each $k \in V$ is a *Nash equilibrium* (NE) of $\Gamma$ if

$$u_k(\mathbf{p}) \geq u_k(x_k, \mathbf{p}_{-k}), \quad \text{for all } x_k \in \Delta_k, \text{ for all } k \in V, \tag{NE}$$

i.e., if there exist no profitable unilateral deviations. By linearity, it suffices to verify the condition in equation (NE) only for pure actions $s_k \in S_k$ instead of all $x_k \in \Delta_k$.

## 3 Joint Learning Model: Q-Learning Dynamics

We next discuss how we can get to the Q-learning dynamics from Q-learning agents when there are multiple learners in the system. Our goal is to identify the dynamics in competitive systems in which multiple agents are playing a rescaled zero-sum polymatrix game repeatedly over time.

**Q-learning.** Q-learning [55, 54] is a value-iteration method for solving the optimal strategies in Markov Decision Processes (MDPs). It can be used as a model where users learn about their optimal strategy when facing uncertainties. Specifically, at every time-point $n \geq 0$, each Q-learning agent, $k \in V$, keeps track of the past performance of each of their actions, $i \in S_k$, via a *Q-value*, denoted by $Q_{ik}(n)$. $Q_{ki}(n)$ is also called the *memory* of agent $k$ about the performance of action $i \in S_k$ up to time step $n \geq 0$. After selecting action $i$ at time $n$, the corresponding Q-value is updated according to the *Q-learning update* rule

$$Q_{ki}(n + 1) = Q_{ki}(n) + \alpha_k \left[ r_{ki}(\mathbf{x}_{-k}, n) - Q_{ki}(n) \right], \ i \in S_k, \tag{4}$$

where $\alpha_k \in [0, 1]$ denotes the learning rate or memory decay of agent $k$ [48, 31]. Agent $k \in V$ updates their actions (choice distributions) according to a *Boltzmann-type distribution*, with

$$x_{ki}(n) = \frac{\exp(Q_{ki}(n)/T_k)}{\sum_{j \in S_k} \exp(Q_{kj}(n)/T_k)}, \quad \text{for each } i \in S_k, \tag{5}$$

where $T_k \in [0, +\infty)$ denotes agent $k$'s learning sensitivity or adaptation, i.e., how much the choice distribution is affected by the past performance. We will refer to $T_k$ as the *exploration rate* or *temperature* of player $k$ [34, 51] (see also next paragraph for a discussion). Combining equations (4) and (5), one obtains the recursive equation of player $k$'s mixed strategy (or choice distribution)

$$x_{ki}(n + 1) = \frac{x_{ki}(n) \exp((Q_{ki}(n + 1) - Q_{ki}(n))/T_k)}{\sum_{j \in S_k} x_{kj}(n) \exp((Q_{kj}(n + 1) - Q_{kj}(n))/T_k)},$$

for each $i \in S_k$. In practice, agents perform large numbers of action updates (updates of Q-values) for each choice-distribution update. This motivates to consider a continuous time version of the learning process of each agent which results in the following update rules for both the memories $Q_{ki}$ and the selection probabilities $x_{ki}$ of each action $i \in S_k$

$$\dot{Q}_{ki} = \alpha_k \left[ r_{ki}(\mathbf{x}_{-k}) - Q_{ki} \right], \quad \text{and} \quad \dot{x}_{ki} = x_{ki} \left( \dot{Q}_{ki} - \sum\nolimits_{j \in S_k} \dot{Q}_{kj} \right) / T_k,$$

where we omitted the dependence on the continuous time, $t \geq 0$. Combining these two expressions with equation (5) under the assumptions that pairs of actions have constant relationship over time and that agents' choice distributions are independently distributed, we obtain the *Q-Learning Dynamics*

$$\dot{x}_{ki} = x_{ki} \left[ r_{ki}(\mathbf{x}_{-k}) - \mathbf{x}_k^\top r_k(\mathbf{x}_{-k}) - T_k \left( \ln(x_{ki}) - \mathbf{x}_k^\top \ln(\mathbf{x}_k) \right) \right], \qquad \text{(QLD)}$$

for all players $k \in V$ and all actions $i \in S_k$.

**Fixed Points of QLD.** Using the convention $x \ln x := 0$ if $x = 0$ (recall that $\lim_{x \to 0^+} x \ln x = 0$), we can see from equation (QLD) that $x_{ki}(t) = x_{ki}(0)$ for all $t > 0$, for all $i \in S_k$ such that $x_{ki}(0) \in \{0, 1\}$. In other words, the boundary of $\Delta_k$ is invariant for (QLD), i.e., if the dynamics start on the boundary, then they will remain there. This implies that all boundary points of the simplex, $\Delta_k$, are trivially fixed points of (QLD).

The interesting part concerns the fixed-points of the dynamics for *interior* starting points. In this case, equation (QLD), implies that, whenever $T_k > 0$, any resulting fixed point of (QLD) can only lie again in the interior of $\Delta_k$ due to the entropy term (ln of the choice probabilities). Clearly, the rate of change of $x_{ki}$ must be equal to zero. Formally, this implies that a strategy profile $\mathbf{p} = (p_k)_{k \in V}$ with $\mathbf{p}_k = (p_{ki})_{i \in S_k}$ for each $k \in V$ is an *interior fixed point* of (QLD) if

$$r_{ki}(\mathbf{p}_{-k}) = \mathbf{p}_k^\top r_k(\mathbf{p}_{-k}) + T_k \left( \ln(p_{ki}) - \mathbf{p}_k^\top \ln(\mathbf{p}_k) \right), \quad \text{for all } k \in V. \qquad \text{(6)}$$

**Exploration Rates.** Equation (QLD) suggests that higher values of $T_k$'s indicate higher *exploration* rates, whereas values close to $0$ indicate a higher *exploitation* rates. Specifically, for $T_k = 0$, the dynamics in (QLD) reduce to the well-known *replicator dynamics*, and agent $k$ simply selects the action $i \in S_k$ with the highest Q-value. On the other hand, as $T_k \to \infty$, player $k$ essentially ignores the input (rewards) from the environment and maximizes the entropy of their current choice distribution (which leads to uniform randomization over their available actions in $S_k$). From a behavioral perspective, this leads to an equivalent interpretation of the $T_k$'s as the *degree of bounded rationality* of the agent [25, 24, 21]. Finally, from an algorithmic perspective, $T_k$ and the corresponding entropy term can be viewed as regularizers that prevent the learning dynamics from overfitting, i.e., reaching the boundary or getting trapped in local optima [10, 16, 38] and [29, 30].

**Exponential Discounting.** In continuous time, the Q-learning updates of equation (4) can be intuitively interpreted as exponential discounting of payoffs, with payoffs further back in the past being less important. In other words, an agent's Q-value for an action shows how beneficial is the action for that agent when payoffs are discounted.

**Proposition 3.1** (Q-Value Updates and Exponential Discounting). *For an agent $k \in V$, consider the Q-learning updates in equation (4) and assume that $Q_i(0) = 0$ for all $i \in S_k$. Then, in continuous time, the Q-value updates are given by*

$$Q_{ik}(t) = \alpha \int_0^t e^{-\alpha_k s} r_i(t - s) ds = \alpha e^{-\alpha_k t} \int_0^t e^{\alpha_k s} r_i(s) \, ds, \quad \text{for any } t > 0.$$

This interpretation also showcases the connection between (QLD) and another commonly studied dynamic in behavioral game theory, Experienced Weighted Attraction (EWA) [14, 21]. Along with the proof of Proposition 3.1, this is explained in more detail in Appendix A.

### 3.1 Solution Concept: Quantal Response Equilibria

When considering agents with bounded rationality, the standard solution concept is the quantal response equilibrium [36]. In its most common definition, a strategy profile $\mathbf{p} = (p_k)_{k \in V}$ with

$\mathbf{p}_k = (p_{ki})_{i \in S_k}$ for each $k \in V$ is an *Quantal Response Equilibrium* (QRE) of $\Gamma$, if it satisfies the following *logit* or *softmax* expression,

$$p_{ki} = \frac{\exp\left(r_{ki}(\mathbf{p}_{-k})/T_k\right)}{\sum_{j \in S_k} \exp\left(r_{kj}(\mathbf{p}_{-k})/T_k\right)}, \quad \text{for all } k \in V, i \in S_k. \tag{QRE}$$

The following property suggests that the interior fixed points of (QLD) coincide with the QRE of $\Gamma$. Following directly from equations (6) and (QRE), this statement will be the starting point of our convergence analysis of (QLD).

**Proposition 3.2** (Interior Fixed Points of (QLD) and QRE)**.** *Let $\Gamma$ be an arbitrary game, with positive exploration rates, $T_k > 0$ for all $k \in V$. Then, for any interior starting point $\mathbf{x}(0) \in \Delta$, the fixed points of the associated* (QLD) *always exist and coincide with the QRE of $\Gamma$. Moreover, given any such fixed point $\mathbf{p}$, we have, for all $\mathbf{x}_k \in \Delta_k$ and for all $k \in V$, that*

$$(\mathbf{x}_k - \mathbf{p}_k)^\top \left[r_k(\mathbf{p}_{-k}) - T_k \ln(\mathbf{p}_k)\right] = 0. \tag{7}$$

Proposition 3.2 does not make use of the weighted network zero-sum property, cf. (2), and thus, it holds for arbitrary network games. We also remark that the QREs are not a refinement of the NEs of $\Gamma$ and, in fact, the two may be significantly different from each other. However, as we argued above, when $T_k = 0$ for all $k \in V$, (QLD) reduce to the standard replicator dynamics and in this case, the QRE coincide with the NE of $\Gamma$, cf. equations (NE) and (QRE).

**Technical Remark.** The condition in equation (7) in Proposition 3.2 is critical for the main step in our proof of convergence of (QLD) in weighted zero-sum polymatrix games. Thus, it is important to note that (7) *only* holds when *all* exploration rates, $T_k, k \in V$ are positive and not merely non-negative. The intuition is that when all $T_k > 0$, every QRE becomes interior, i.e., has full support. This implies that all deviations to pure or mixed actions give the *same* expected reward to the deviating agent. If some $T_k$ are equal to 0, then (7) holds with (non-strict) inequality which is not generally enough for the proof of Theorem 4.1. This case exhibits more interesting behavior and will be discussed in detail in the experimental section. In brief, if this is the case, then the dynamics may also converge to a boundary point even if the dynamics start in the interior.

## 4 Convergence of Q-learning in Weighted Zero-sum Polymatrix Games

Our main result is that Q-learning (QLD) converges to the QRE of any weighted zero-sum polymatrix game, $\Gamma$. Importantly, when the exploration rates of all agents are positive, this QRE is unique. The key step in the proof of both claims is to show that the *distance* between an interior QRE, $\mathbf{p} \in \Delta$, and the sequence of play, $\mathbf{x}(t), t \geq 0 \in \Delta$, that is generated by (QLD) is monotonically decreasing. To measure this distance in a meaningful way, we will use the notion of KL-divergence which is formally defined next.

**Definition 1** (Kullback-Leibler (KL) Divergence)**.** The Kullback-Leibler or *KL-Divergence* (also called *relative entropy*), $D_{\mathrm{KL}}$, between two strategy profiles $\mathbf{p} = (\mathbf{p}_k)_{k \in V}, \mathbf{x}(t) = (\mathbf{x}_k(t))_{k \in V} \in \Delta$ with $\mathbf{p}_k = (p_{ki})_{i \in S_k}$ and $\mathbf{x}_k(t) = (x_{ki}(t))_{i \in S_k} \in \Delta_k$ for all $k \in V$, is defined as

$$D_{\mathrm{KL}}\left(\mathbf{p} \parallel \mathbf{x}(t)\right) := \sum_{k \in V} D_{\mathrm{KL}}(\mathbf{p}_k \parallel \mathbf{x}_k(t)) = \sum_{k \in V} \mathbf{p}_k^\top \ln\left(\frac{\mathbf{p}_k}{\mathbf{x}_k(t)}\right), \tag{8}$$

where $\mathbf{p}_k^\top \ln\left(\frac{\mathbf{p}_k}{\mathbf{x}_k(t)}\right) = \sum_{i \in S_k} p_{ki} \ln\left(p_{ki}/x_{ki}(t)\right)$. If $\mathbf{w} = (w_k)_{k \in V}$ is a $k$-dimensional vector of positive scalars, then the *weighted or rescaled KL-divergence*, $D_{\mathrm{KL}(w)}$, is defined as

$$D_{\mathrm{KL}(w)}\left(\mathbf{p} \parallel \mathbf{x}(t)\right) := \sum_{k \in V} w_k D_{\mathrm{KL}}(\mathbf{p}_k \parallel \mathbf{x}_k(t)). \tag{9}$$

The KL-divergence between $\mathbf{p}$ and $\mathbf{x}(t)$ can be thought of as a measurement of how far the distribution $\mathbf{p}$ is from the distribution $\mathbf{x}(t)$. However, the KL-divergence is not symmetric, i.e., in general it holds that $D_{\mathrm{KL}}(\mathbf{p} \parallel \mathbf{x}(t)) \neq D_{\mathrm{KL}}(\mathbf{x}(t) \parallel \mathbf{p})$. Using the notion of KL-divergence we can now formulate our main result.

**Theorem 4.1** (Convergence of (QLD) to (QRE)). *Let $\Gamma$ be a rescaled zero-sum polymatrix game, with positive exploration rates $T_k$. There exists a unique QRE $\mathbf{p}$ such that if $\mathbf{x}(t), t \geq 0$ is any trajectory of the associated Q-learning dynamics, $\dot{\mathbf{x}} = f(\mathbf{x})$, where $f_i$ is given via (QLD), and $\mathbf{x}(0)$ is an interior point, then $\mathbf{x}(t)$ converges to $\mathbf{p}$ exponentially fast. In particular, we have that*

$$\frac{d}{dt} D_{KL(w)}(\mathbf{p} \parallel \mathbf{x}(t)) = -\sum\nolimits_{k \in V} w_k T_k \left[ D_{KL}(\mathbf{p}_k \parallel \mathbf{x}_k) + D_{KL}(\mathbf{x}_k \parallel \mathbf{p}_k) \right]. \tag{10}$$

*Sketch of Proof.* Let $\mathbf{p} = (\mathbf{p}_k)$ be QRE of $\Gamma$ (existence of $\mathbf{p}$ is guaranteed via Proposition 3.2), and let $\mathbf{x}(t)$ be a trajectory of the Q-learning dynamics (QLD) with $\mathbf{x}(0)$ an interior point. We will first establish (10), from which the other statements follow. The first step is to derive an explicit formula for the time derivative of the KL divergence between $\mathbf{p}_k$ and $\mathbf{x}_k(t)$ for a given player $k$.

**Lemma 4.2.** *For any QRE equilibrium $\mathbf{p}$ and any player $k \in V$ we have that*

$$\frac{d}{dt} D_{KL}(\mathbf{p}_k \parallel \mathbf{x}_k(t)) = (\mathbf{x}_k - \mathbf{p}_k)^\top \left[ r_k(\mathbf{x}_{-k}) - r_k(\mathbf{p}_{-k}) \right] - T_k \left[ D_{KL}(\mathbf{p}_k \parallel \mathbf{x}_k) + D_{KL}(\mathbf{x}_k \parallel \mathbf{p}_k) \right].$$

It follows from Lemma 4.2 that

$$\frac{d}{dt} D_{\text{KL}}(\mathbf{p} \parallel \mathbf{x}(t)) = \sum\nolimits_{k \in V} \frac{d}{dt} w_k D_{\text{KL}}(\mathbf{p}_k \parallel \mathbf{x}_k)$$

$$= \sum_{k \in V} w_k \left[ (\mathbf{x}_k - \mathbf{p}_k)^\top \left[ r_k(\mathbf{x}_{-k}) - r_k(\mathbf{p}_{-k}) \right] - T_k \left( D_{\text{KL}}(\mathbf{p}_k \parallel \mathbf{x}_k) + D_{\text{KL}}(\mathbf{x}_k \parallel \mathbf{p}_k) \right) \right] \tag{11}$$

$$= \sum_{k \in V} w_k (\mathbf{x}_k - \mathbf{p}_k)^\top \left[ r_k(\mathbf{x}_{-k}) - r_k(\mathbf{p}_{-k}) \right] - \sum_{k \in V} w_k T_k \left[ D_{\text{KL}}(\mathbf{p}_k \parallel \mathbf{x}_k) + D_{\text{KL}}(\mathbf{x}_k \parallel \mathbf{p}_k) \right].$$

Notice that (10) follows from the last line of (11) if the first term vanishes. This can be shown by utilizing the following result regarding weighted zero-sum polymatrix games.

**Lemma 4.3.** *Given any QRE, $\mathbf{p} = (\mathbf{p}_k)_{k \in V}$, and any point $\mathbf{x} = (\mathbf{x}_k)_{k \in V} \in \Delta$ we have that*

$$\sum\nolimits_{k \in V} w_k \left[ \mathbf{x}_k^\top r_k(\mathbf{p}_{-k}) + \mathbf{p}_k^\top r_k(\mathbf{x}_{-k}) \right] = 0.$$

Returning to (11), we have that the first term in the right-hand side of the last equation vanishes since

$$\sum\nolimits_{k \in V} w_k (\mathbf{x}_k - \mathbf{p}_k)^\top \left[ r_k(\mathbf{x}_{-k}) - r_k(\mathbf{p}_{-k}) \right] = \sum\nolimits_{k \in V} w_k \mathbf{x}_k^\top r_k(\mathbf{x}_{-k}) + \sum_{k \in V} w_k \mathbf{p}_k^\top r_k(\mathbf{p}_{-k})$$

$$+ \sum\nolimits_{k \in V} w_k \left[ \mathbf{x}_k^\top r_k(\mathbf{p}_{-k}) + \mathbf{p}_k^\top r_k(\mathbf{x}_{-k}) \right] = 0,$$

where the last equality follows from both the zero-sum property (2) and Lemma 4.3. Thus,

$$\frac{d}{dt} D_{\text{KL}}(\mathbf{p} \parallel \mathbf{x}(t)) = -\sum\nolimits_{k \in V} w_k T_k \left[ D_{\text{KL}}(\mathbf{p}_k \parallel \mathbf{x}_k) + D_{\text{KL}}(\mathbf{x}_k \parallel \mathbf{p}_k) \right].$$

This equality implies that $D_{\text{KL}(w)}$ is a Lyapunov function for the Q-learning dynamics. Moreover, it implies that $\mathbf{p}$ is unique, and that $\mathbf{x}(t)$ is converging to $\mathbf{p}$ at an exponential rate. $\square$

**Approximate Equilibrium Selection Mechanism.** Theorem 4.1 provides a tractable, algorithmic approach to the problem of equilibrium selection in weighted zero-sum polymatrix games. Given any such a game $\Gamma$, an interior initial condition and positive exploration rates $T_k$, Theorem 4.1 guarantees that the Q-learning agents will converge to a unique QRE of $\Gamma$ at an exponentially fast rate. This unique QRE can be very different from any Nash equilibrium of $\Gamma$ if the exploration rates are high, but it will be arbitrarily close to some Nash equilibrium of $\Gamma$ as exploration rates approach zero (yet remain positive for all agents). Thus, exploration by all agents creates an arbitrarily good and efficient *approximate equilibrium selection mechanism* for the original game.

As discussed above (see Technical Remark in Section 3.1), the condition that the exploration rates of *all* agents are positive is necessary to establish convergence of QLD to a unique QRE of a weighted zero-sum polymatrix game. As demonstrated next in our experimental section, there can be multiple QREs if one or some agents have zero exploration rates.

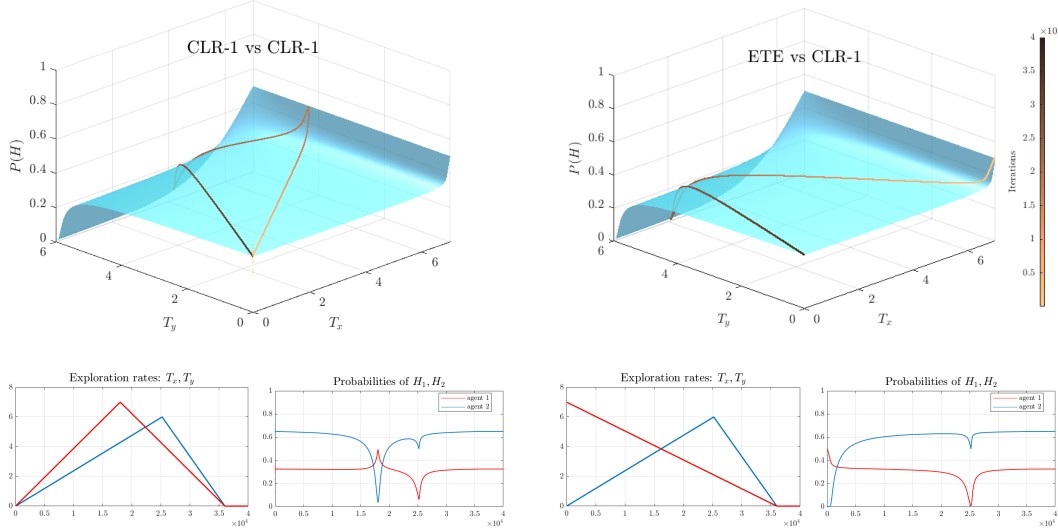

Figure 1: QRE surface and exploration paths (upper tiles) for two different exploration policy profiles (lower tiles) in the Asymmetric MPs game. For any combination of exploration policies (CLR-1 and ETE), the sequence of play converges to the unique QRE and as the exploration rates decrease to zero, the sequence of play converges to the unique Nash equilibrium of the game.

## 5 Experiments: Equilibrium Selection in Competitive Games

**Two-Agent Weighted Zero-Sum Games.** Starting with the performance of (QLD) in this low-dimensional case, we can visualize the QRE surface and exploration paths and gain intuition that carries over to the higher dimensional cases with more agents that we treat later.

**Experimental setup:** We consider the 2-agent *Asymmetric Matching Pennies* (AMPs) game, a variation of the matching pennies [58], in which each agent has two actions, $\{H, T\}$, and payoffs

$$\mathbf{A} = \begin{pmatrix} 2 & -2 \\ 0 & 2 \end{pmatrix} \mathbf{B} = \begin{pmatrix} 4 & 0 \\ -4 & -4 \end{pmatrix}. \tag{AMPs}$$

The AMPs game is a weighted zero-sum game since $\mathbf{A} + 0.5 \cdot \mathbf{B}^\top = 0$ (each agent is the row agent in their matrix), with a unique interior Nash equilibrium, $(\mathbf{p}, \mathbf{q}) = ((1/3, 2/3), (2/3, 1/3))$.

**Results:** In Figure 1, we visualize the QRE surfaces (light blue manifolds) for different exploration rates $T_x, T_y$ of the two agents ($x - y$ plane). The vertical axis shows the probability with which agent 1 chooses $H$ at the unique QRE of the game. We plot the exploration path along two representative exploration-exploitation policies: *Explore-Then-Exploit* (ETE) [5], which starts with (relatively) high exploration that gradually reduces to zero and *Cyclical Learning Rate with 1 cycle* (CLR-1) [50], which starts with low exploration, increases to high exploration around the half-life of the cycle and then decays to 0. For each pair of exploration rates, the learning dynamics converge to the corresponding unique QRE. As the exploration rates decay to zero, the dynamics converge to (i.e., select) the unique QRE (i.e., the Nash equilibrium in this case) of the original game.

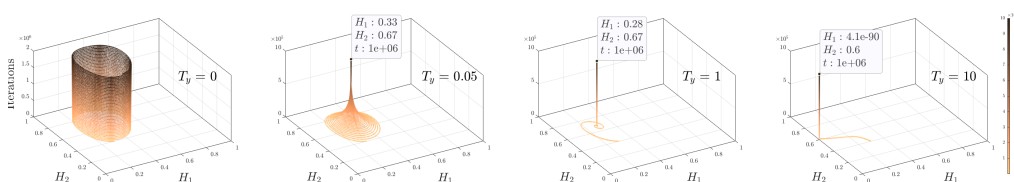

Figure 2: Q-learning dynamics in the AMPs game for $T_x = 0$ (no exploration by $x$-agent) and four different exploration rates, $T_y \geq 0$ by $y$-agent.

By contrast, the first tile of Figure 2 shows that the dynamics cycle around the unique Nash equilibrium when both agents do not use exploration. The rest of the tiles of Figure 2 show that in this case, exploration by only one agent suffices to lead the joint-learning dynamics to converge to a unique QRE. In this case, for higher values of exploration by the exploring agent, the QRE component of the non-exploring agent may lie at the boundary (see tile 4 in Figure 2 and Appendix C).

**Network Zero-Sum Games.** As the next experiment shows, equilibrium selection works also in larger networks, provided that all agents maintain a positive exploration rate. In contrast to the previous example, in this case, exploration by only some agents is not sufficient for convergence.

**Experimental setup:** We consider the *Match-Mismatch* weighted zero-sum polymatrix game (MMG) with $n + 2$ agents ($n \in \mathbb{N}$ is arbitrary) which is shown in Figure 3. Each of the agents $p_1$ to

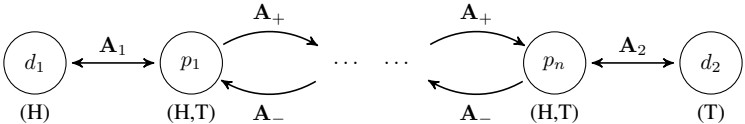

Figure 3: The match-mismatch weighted zero-sum polymatrix game (MMG).

$p_n$ has two actions, $\{H, T\}$, and receives $+1$ if they match the action of the next agent (otherwise they receive $-1$) and $+1$ if they mismatch the strategy of the previous agent (otherwise they receive $-1$). There are two dummy agents, $d_1, d_2$, who have the same payoffs but only one action: H for $d_1$ and T for $d_2$. The payoff matrices for the games between non-dummy agents are given by

$$\mathbf{A}_+ = \begin{pmatrix} 1 & -1 \\ -1 & 1 \end{pmatrix}, \quad \mathbf{A}_- = -\mathbf{A}_+ \tag{MMG}$$

whereas the payoff matrices of dummy agents $d_1$ and $d_2$ are given by $\mathbf{A}_1 = \mathbf{A}_2 = (1, -1)$. The column agents in the payoff matrices $\mathbf{A}_1$ and $\mathbf{A}_2$ are $p_1$ and $p_n$, and their payoffs in the encounters against $d_1$ and $d_2$ are given by $-\mathbf{A}_1^\top$ and $-\mathbf{A}_2^\top$, respectively. This game has infinite many NE of the following form: the odd-numbered agents play strategy $T$ (which, for agent 1 ensures $+1$ against $d_1$) whereas the even agents are indifferent between $H, T$ (since they are certain to have opposing results in the two games against the previous and the next agent, both being odd and playing $T$).

**Results:** In Figure 4, we consider an instance of (MMG) with $n = 3$ non-dummy agents and show the projections (tiles 1 to 3) of the QRE surfaces on the $H$ coordinate at QRE for agents 1 to 3 for fixed $T_3 = 3$ and all possible combinations of $T_1, T_2$ (x-y planes). The QRE manifolds continuously approach the boundary (case with $T_i = 0$ for $i = 1, 2$) which leads to a unique equilibrium selection also when $T_1 = T_2 = 0$. Tile 4 shows a snapshot of the Lyapunov function (KL-divergence between choice distribution $\mathbf{x}$ and unique QRE $\mathbf{q}$) in a 7 non-dummy agents instance of (MMG) for a fixed exploration profile with positive exploration rates for all agents, $T_k > 0$, for all $k = 1, \ldots, 7$. To obtain this plot, (which is similar for all $n > 1$), we use the dimension reduction technique of [35] to visualize high-dimensional surfaces along two randomly chosen directions (cf. Appendix C). As expected, the KL-divergence is convex and decreasing for all $\mathbf{x}$ with a unique minimizer.

Figure 5 shows summary statistics for the $n = 7$ (MMG) instance for three different exploration profiles. The tiles in the left column show the equilibria (top) and utilities (bottom) of the 7 agents

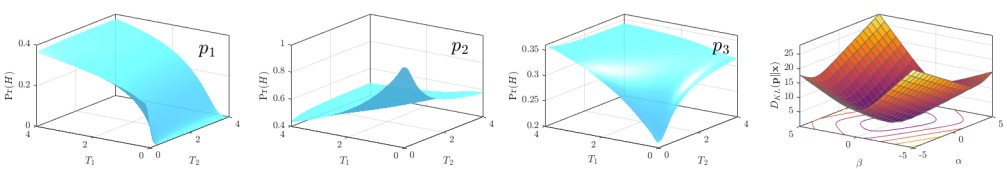

Figure 4: Projections of the QRE surfaces on the $H$ coordinate of players $p_1, p_2, p_3$ (tiles 1-3) and a snapshot of the Lyapunov function, KL-divergence, (tile 4) in two instances of the (MMG) game with 3 and 7 non-dummy agents, respectively, as discussed in Section 5.

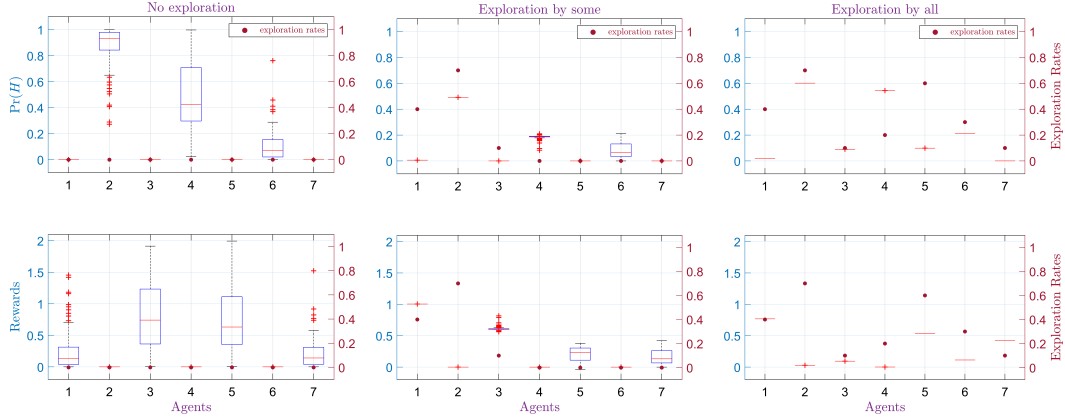

Figure 5: Summary statistics from 100 runs with 3 profiles of exploration rates in a 7 non-dummy agent instance of (MMG). No exploration leads to multiple, payoff-diverse equilibria (left), exploration by some agents is not enough to ensure convergence to a unique outcome which shows the tightness of Theorem 4.1 (middle) and exploration by all agents leads to a unique QRE (right column).

in 100 runs when $T_k = 0$ for all $k = 1, 2, \ldots, n$ (no exploration). In this case, (QLD) converge in all runs to the pure action for the odd agents and to some arbitrary (and possibly different between runs) mixed action for the even agents. This behavior of (QLD) conforms with previous results in adversarial learning concerning equilibria that lie on one face (relative boundary) of the high-dimensional simplex (cf. [38]). The tiles in the middle column correspond to a case with exploration by agents $1, 2, 3$ and no exploration by agents $4$ to $7$. Exploration by the first group of agents ensures convergence of their individual dynamics to the corresponding component of the QRE but even-numbered agents that lie further away from that group fail to converge to a unique outcome or to obtain a unique payoff (boxplots for agents 3 to 7). The outcome in this case, shows that the statement of Theorem 4.1 is tight, and exploration by only a subset of agents may not be enough to ensure a unique outcome in general settings. In particular, zero exploration by agents 4 to 7 leads to multiple equilibria for agents 4 and 6 (upper middle tile) and to non-unique payoffs for agents 5 and 7 but also for agent 3 who has a positive exploration rate (lower middle tile). Finally, the tiles in the right column show convergence to a unique QRE when all agents have positive exploration rates.

## 6   Conclusions

In this paper, we studied a commonly used smooth variant of the Q-learning dynamics in multi-agent competitive games. Given that each agent's strategy includes at least some amount of exploration, we showed that the dynamics always converge to a unique Quantal Response Equilibrium (QRE) of the game at an exponentially fast rate. The convergence of Q-learning in this (competitive) setting is a remarkably robust phenomenon, occurring regardless of the number of agents or degree of competition, and without any need for parameter fine-tuning.

Our theoretical results have important implications from an algorithmic perspective. In contrast to two-agent zero-sum games in which agents have the same reward in all equilibria, multi-agent systems typically exhibit a multiplicity of equilibria in which each agent may have different rewards. This inherent complexity within multi-agent systems leads to the non-trivial problem of equilibrium selection which has been previously studied only within *cooperative* settings. The exponentially fast convergence of the Q-learning dynamics to a unique QRE that we prove in this paper establishes an efficient and simple mechanism to select (approximate) Nash equilibria, implying that the problem of equilibrium selection is tractable within competitive multi-agent settings as well. In particular, while a QRE may correspond to a significantly different outcome from all Nash equilibria of the original game when exploration rates are high, it offers an arbitrarily good approximation of some Nash equilibrium for close-to-zero exploration rates. Together with [34] our results establishe that exploration-exploitation with Q-learning works well in both as an equilibrium selection mechanism in both cooperative and competitive settings and provide a solid starting point for the study of general, *mixed* multi-agent settings that involve both cooperative and competitive interactions.

## Acknowledgments and Disclosure of Funding

This research/project is supported in part by the National Research Foundation, Singapore under its AI Singapore Program (AISG Award No: AISG2-RP-2020-016), NRF 2018 Fellowship NRF-NRFF2018-07, NRF2019-NRF-ANR095 ALIAS grant, grant PIE-SGP-AI-2018-01, AME Programmatic Fund (Grant No. A20H6b0151) from the Agency for Science, Technology and Research (A*STAR), and by EPSRC grant EP/R018472/1.

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
