# Supplementary Material
# Exploration-Exploitation in Multi-Agent Competition: Convergence with Bounded Rationality

## A   Omitted Proofs and Materials: Section 3

We first restate and prove Proposition 3.1.

**Proposition 3.1.** *Consider the Q-learning updates*

$$Q_i(n+1) = Q_i(n) + \alpha[r_i(n) - Q_i(n)] \tag{12}$$

*where $n \geq 0$ are discrete time steps and assume that $Q_i(0) = 0$ for all $i \in A$. Then, in continuous time, the updates are given by*

$$Q_i(t) = \alpha \int_0^t e^{-\alpha s} r_i(t-s) ds = \alpha e^{-\alpha t} \int_0^t e^{\alpha s} r_i(s)\, ds, \quad \text{for any } t > 0.$$

*Proof.* We will provide two proofs of the above statement. For the first, we solve the recursion in equation (12) and then use the approximation of the exponential function by the geometric sum. Specifically, we have that

$$\begin{aligned}
Q_i(n+1) &= \alpha r_i(n) + (1-\alpha)Q_i(n) \\
&= \alpha r_i(n) + (1-\alpha)[\alpha r_i(n-1) + (1-\alpha)Q_i(n-1)] \\
&= \alpha r_i(n) + (1-\alpha)\alpha r_i(n-1) + (1-\alpha)^2 Q_i(n-1) \\
&= \alpha r_i(n) + (1-\alpha)\alpha r_i(n-1) + (1-\alpha)^2[\alpha r_i(n-2) + (1-\alpha)Q_i(n-2)] \\
&= \dots \\
&= (1-\alpha)^{n+1} Q_i(0) + \sum_{k=0}^{n} \alpha(1-\alpha)^k r_i(n-k) \\
&= \sum_{k=0}^{n} \alpha(1-\alpha)^k r_i(n-k)
\end{aligned}$$

where in the last equation we did a change of variables in the summation since $Q_i(0) = 0$ by assumption. By taking continuous time steps (i.e., if instead of $k \to k+1$, we consider $k \to \Delta k$ with $\Delta k \to 0$), then the above becomes

$$Q(t) = \alpha \int_0^t e^{-as} r_i(t-s) ds$$

as claimed. By a change of variables, we also obtain the second expression in the Lemma's statement. The second way to prove this statement is by directly considering a continuous time version of (12). In this case, we have that

$$\dot{Q}_i(s) = \alpha[r_i(s) - Q_i(s)],$$

where $\dot{Q}_i(s)$ denotes the time derivative of $Q_i(s)$, i.e., $dQ_i(s)/ds$. This is a first order non-homogeneous (in the constant term) linear differential equation of the form

$$\dot{Q}_i(s) + \alpha Q_i(s) = \alpha r_i(s).$$

By multiplying both sides with the integrating factor $e^{\alpha s}$, we obtain that

$$e^{\alpha s}[\dot{Q}_i(s) + \alpha Q_i(s)] = \alpha e^{\alpha s} r_i(s) \implies \frac{d}{ds}(e^{\alpha s} Q_i(s)) = \alpha e^{\alpha s} r_i(s).$$

35th Conference on Neural Information Processing Systems (NeurIPS 2021).

We can now integrate both sides of the last expression from $0$ to $t$ to obtain that

$$e^{\alpha t}Q_i(t) = \alpha \int_0^t e^{\alpha s} r_i(s)ds \implies Q_i(t) = \alpha e^{-\alpha t}\int_0^t e^{\alpha s} r_i(s)ds$$

as claimed, where the integration constant disappears due to the boundary condition $Q_i(0) = 0$. $\quad\square$

## A.1   Derivation of Q-learning dynamics

We next provide two formal derivations of the Q-learning dynamics of equation (QLD). Similar calculations can be found in [51, 48, 47, 31] and [21, 42].

**Proposition A.1** (Derivation of Q-learning Dynamics). *Assume that agents select their actions according to a Boltzmann (or softmax) distribution with parameter $T$, i.e.,*

$$x_i(t) = \frac{\exp\left(Q_i(t)/T\right)}{\sum_{j\in A}\exp\left(Q_j(t)/T\right)} \quad \text{for any } t > 0, \tag{13}$$

*with $x_i(0)$ is initialized arbitrarily for all $i \in A$, where $Q_i(t)$ is the Q-value of action $i$ at time $t$. Then, the evolution of the action probabilities is governed by the Q-learning dynamics*

$$\dot{x}_i(t) = x_i(t)\left[r_i(t) - \sum_{j\in A} r_j(t)x_j(t) - T\left(\ln x_i(t) - \sum_{j\in A} x_j(t)\ln x_j(t)\right)\right]. \tag{14}$$

*Proof.* The time derivative of $x_i(t)$ is equal to

$$\dot{x}_i(t) = \frac{\exp\left(Q_i(t)/T\right)}{T\sum_{j\in A}\exp\left(Q_j(t)/T\right)}\left[\dot{Q}_i(t) - \frac{\sum_{j\in A}\dot{Q}_j(t)\exp\left(Q_j(t)/T\right)}{\sum_{j\in A}\exp\left(Q_j(t)/T\right)}\right]$$

$$= \frac{1}{T}x_i(t)\left[\dot{Q}_i(t) - \sum_{j\in A}\left(\frac{\exp\left(Q_j(t)/T\right)}{\sum_{j\in A}\exp\left(Q_j(t)/T\right)}\right)\dot{Q}_j(t)\right]$$

$$= \frac{1}{T}x_i(t)\left[\dot{Q}_i(t) - \sum_{j\in A} x_j(t)\dot{Q}_j(t)\right].$$

Hence, using that $\dot{Q}_i(t) = \alpha[r_i(t) - Q_i(t)]$, the above yields

$$\dot{x}_i(t) = \frac{\alpha}{T}x_i(t)\left[r_i(t) - Q_i(t) - \sum_{j\in A} x_j(t)[r_j(t) - Q_j(t)]\right]$$

$$= \frac{\alpha}{T}x_i(t)\left[r_i(t) - \sum_{j\in A} x_j(t)r_j(t) - \left(Q_i(t) - \sum_{j\in A} x_j(t)Q_j(t)\right)\right]. \tag{$*$}$$

Finally, taking $\ln$ in equation (13), and solving for $Q_i(t)$, we have that

$$Q_i(t) = T\left[\ln x_i(t) + \ln\left(\sum_{j\in A}\exp\left(Q_j(t)/T\right)\right)\right]$$

In the above expression, we may set $C := \ln\left(\sum_{j\in A}\exp\left(Q_j(t)/T\right)\right)$ since this term is the same for all $i \in A$. Thus, we have that

$$Q_i(t) - \sum_{j\in A} x_j(t)Q_j(t) = T\left(\ln x_i(t) + C\right) - \sum_{j\in A} x_j(t)T\left(\ln x_i(t) + C\right)$$

$$= T\left(\ln x_i(t) - \sum_{j\in A} x_j(t)\ln x_j(t)\right) + T\left(C - \sum_{j\in A} x_j(t)C\right)$$

$$= T\left(\ln x_i(t) - \sum_{j\in A} x_j(t)\ln x_j(t)\right)$$

since $\sum_{j \in A} x_j(t) = 1$ and hence, the terms $C$ and $\sum_{j \in A} x_j(t)C$ cancel out. Substituting the last expression in equation $(*)$, we obtain the desired solution, namely

$$\dot{x}_i(t) = \frac{\alpha}{T} x_i(t) \left[ r_i(t) - \sum_{j \in A} x_j(t) r_j(t) - T \left( \ln x_i(t) - \sum_{j \in A} x_j(t) \ln x_j(t) \right) \right].$$

Rescaling time by $t \to \alpha t / T$, we obtain the solution. $\qquad\square$

**An Alternative Derivation** (see also [21, 42]). Another way to obtain the Q-learning dynamics in equation (14) is by a direct substitution of (12) in (13). In this case, we have that

$$x_i(t+1) = \frac{\exp\left(Q_i(t+1)/T\right)}{\sum_{j \in A} \exp\left(Q_j(t+1)/T\right)}$$

$$= \frac{\exp\left((\alpha r_i(t+1) + (1-\alpha)Q_i(t))/T\right)}{\sum_{j \in A} \exp\left((\alpha r_j(t+1) + (1-\alpha)Q_j(t))/T\right)}$$

$$= \frac{\exp\left((1-\alpha)Q_i(t)/T\right) \cdot \exp\left(\alpha r_i(t+1)/T\right)}{\sum_{j \in A} \exp\left((1-\alpha)Q_j(t)/T\right) \cdot \exp\left(\alpha r_j(t+1)/T\right)}$$

$$= \frac{\left(\exp\left(Q_i(t)/T\right)\right)^{(1-\alpha)} \cdot \exp\left(\alpha r_i(t+1)/T\right)}{\sum_{j \in A} \left(\exp\left(Q_j(t)/T\right)\right)^{(1-\alpha)} \cdot \exp\left(\alpha r_j(t+1)/T\right)}$$

$$= \frac{x_i(t)^{(1-\alpha)} \cdot \exp\left(\alpha r_i(t+1)/T\right)}{\sum_{j \in A} x_j(t)^{(1-\alpha)} \cdot \exp\left(\alpha r_j(t+1)/T\right)}$$

where the last equality is obtained by dividing both numerator and denominator with the normalizing constant $\sum_{k \in A} \left(\exp\left(Q_k(t)/T\right)\right)^{(1-\alpha)}$. By taking $\ln$ of both sides in the previous equation, we then have that

$$\ln x_i(t+1) = \ln x_i(t) - \alpha \ln x_i(t) + \frac{\alpha}{T} r_i(t+1) - \ln\left(\sum_{j \in A} x_j(t)^{(1-\alpha)} \cdot \exp\left(\alpha r_j(t+1)/T\right)\right)$$

or equivalently

$$\ln x_i(t+1) - \ln x_i(t) = \frac{\alpha}{T} r_i(t+1) - \alpha \ln x_i(t) - \ln C$$

where $C := \sum_{j \in A} x_j(t)^{(1-\alpha)} \cdot \exp\left(\alpha r_j(t+1)/T\right)$ is the denominator of the previous expression which is the same for all $i \in A$. Thus, in continuous time, the above equation becomes

$$\frac{d}{dt} \ln x_i(t) = \frac{\alpha}{T} r_i(t) - \alpha \ln x_i(t) - \ln C$$

which yields

$$\dot{x}_i(t) = \frac{\alpha}{T} x_i(t) \left[ r_i(t) - T \ln x_i(t) - \frac{T}{\alpha} \ln C \right]. \qquad (**)$$

To determine $\ln C$, note that $\sum_{i \in A} \dot{x}_i(t) = 0$ since $\sum_{i \in A} x_i(t) = 1$ for all $t$ (i.e., the sum of the $x_i$'s remains constant, and equal to 1, at all times $t \geq 0$). Thus, summing over all $i \in A$, we obtain

$$0 = \sum_{i \in A} x_i(t) r_i(t) - T \sum_{i \in A} x_i(t) \ln x_i(t) - \frac{T}{\alpha} \ln C \sum_{i \in A} x_i(t)$$

$$= \sum_{i \in A} x_i(t) r_i(t) - T \sum_{i \in A} x_i(t) \ln x_i(t) - \frac{T}{\alpha} \ln C$$

or equivalently

$$\frac{T}{\alpha} \ln C = \sum_{i \in A} x_i(t) r_i(t) - T \sum_{i \in A} x_i(t) \ln x_i(t)$$

Substituting this expression back in equation $(**)$, we obtain

$$\dot{x}_i(t) = \frac{\alpha}{T} x_i(t) \left[ r_i(t) - T \ln x_i(t) - \sum_{j \in A} x_j(t) r_j(t) + T \sum_{j \in A} x_j(t) \ln x_j(t) \right]$$

$$= \frac{\alpha}{T} x_i(t) \left[ r_i(t) - \sum_{j \in A} x_j(t) r_j(t) - T \left( \ln x_i(t) - \sum_{i \in A} x_j(t) \ln x_j(t) \right) \right],$$

which after rescaling time to $t \to \alpha t / T$ is precisely the expression of the Q-learning dynamics in equation (14).

**Relation to Experience Weighted Attraction (EWA) Learning**    (see also [42]). In Experience Weighted Learning (EWA), the *attractions* (the equivalent of Q-values), $Q_i(t)$ of each action $i \in A$ are updated according to the following scheme

$$Q_i(t+1) = \frac{(1-\alpha)N(t)Q_i(t) + [\delta + (1-\delta)I(i, s_i(t))]r_i(t)}{N(t+1)},$$

$$N(t+1) = \rho N(t) + 1,$$

where $I(i, s(t)) = 1$ if $i = s(t)$ and 0 otherwise. Here $s(t)$ denotes the action taken by the agent at time $t$. The variables $N(t)$ and $Q_i(t)$ are initialized arbitrarily at $t = 0$, but typically, they are set to be 0 at $t = 0$. For $\rho = 0$, i.e., in the case in which previous experience does not reduce the impact of current rewards, the above system becomes

$$Q_i(t+1) = (1-\alpha)Q_i(t) + \begin{cases} r_i(t), & \text{if } s(t) = i, \\ \delta r_i(t), & \text{otherwise} \end{cases}$$

Thus, for $\delta = 1$, the EWA update rule becomes equal to the Q-learning updates, up to a constant $\alpha$ in the rewards.

## A.2    Q-learning and Quantal Response Equilibria

We next restate and prove Proposition 3.2.

**Proposition 3.2.** *Let $\Gamma$ be an arbitrary game, with positive exploration rates $T_k$ and consider the associated Q-learning dynamics*

$$\dot{x}_{ki} = x_{ki} \left[ r_{ki}(\mathbf{x}_{-k}) - \mathbf{x}_k^\top r_k(\mathbf{x}_{-k}) - T_k \left( \ln(x_{ki}) - \mathbf{x}_k^\top \ln(\mathbf{x}_k) \right) \right], \quad i \in S_k, k \in V.$$

*The interior fixed points, $\mathbf{p} = (p_k)_{k \in V}$ of the Q-learning dynamics are the solutions of the system*

$$p_{ki} = \frac{\exp(r_{ki}(\mathbf{p}_{-k})/T_k)}{\sum_{j \in S_k} \exp(r_{kj}(\mathbf{p}_{-k})/T_k)}, \quad \text{for all } i \in S_k. \tag{15}$$

*Such fixed points always exist and coincide with the Quantal Response Equilibria (QRE) of $\Gamma$. Given any such fixed point $\mathbf{p}$, we have, for all $\mathbf{x}_k \in \Delta_k$ and for all $k \in V$, that*

$$(\mathbf{x}_k - \mathbf{p}_k)^\top [r_k(\mathbf{p}_{-k}) - T_k \ln(\mathbf{p}_k)] = 0. \tag{16}$$

*Proof.* Solving equation (6) for $\ln q_{ki}$ and applying the exponential function on both sides of the resulting equation yields that

$$p_{ki} = \exp\left( \frac{r_{ki}(\mathbf{p}_{-k}) - \mathbf{p}_k^\top r_k(\mathbf{p}_{-k}) + T_k \mathbf{p}_k^\top \ln(\mathbf{p}_k)}{T_k} \right)$$

$$= \exp\left( \frac{r_{ki}(\mathbf{p}_{-k})}{T_k} \right) \cdot \exp\left( \frac{-\mathbf{p}_k^\top r_k(\mathbf{p}_{-k}) + T_k \mathbf{p}_k^\top \ln(\mathbf{p}_k)}{T_k} \right).$$

The second term in the last equation, i.e., $\exp\left( \frac{-\mathbf{p}_k^\top r_k(\mathbf{p}_{-k}) + T_k \mathbf{p}_k^\top \ln(\mathbf{p}_k)}{T_k} \right)$, is the same (constant) for all $i \in S_k$. Thus, denoting this term by $Z$, we have that

$$p_{ki} = \exp\left( \frac{r_{ki}(\mathbf{p}_{-k})}{T_k} \right) \cdot Z, \quad \text{for all } i \in S_k.$$

Since $\mathbf{p}_k$ lies in $\Delta_k$, it must be the case that $\sum_{j \in S_k} p_{ki} = 1$, which implies that

$$\sum_{i \in S_k} p_{ki} = \sum_{i \in S_k} \exp\left(\frac{r_{ki}(\mathbf{p}_{-k})}{T_k}\right) \cdot Z = 1 \implies Z = \left(\sum_{i \in S_k} \exp\left(\frac{r_{ki}(\mathbf{p}_{-k})}{T_k}\right)\right)^{-1}.$$

Substituting back in the expression for $p_{ki}$, we obtain that

$$p_{ki} = \frac{\exp\left(r_{ki}(\mathbf{p}_{-k})/T_k\right)}{\sum_{j \in S_k} \exp\left(r_{kj}(\mathbf{p}_{-k})T_k\right)},$$

as claimed in equation (15). Existence follows from the application of Brouwer's fixed point theorem on the continuous map defined by the previous in $\Delta_k$, see [36]. To obtain equation (16), we observe that for each $x_k \in \Delta_k$ and each $k \in V$, equation (6) implies that

$$\mathbf{x}_k^\top r_k(\mathbf{p}_{-k}) = \mathbf{p}_k^\top r_k(\mathbf{p}_{-k}) - T_k(\mathbf{p}_k - \mathbf{x}_k)^\top \ln(\mathbf{p}_k), \quad \text{for all } k \in V.$$

or equivalently that

$$(\mathbf{x}_k - \mathbf{p}_k)^\top [r_k(\mathbf{p}_{-k}) - T_k \ln(\mathbf{p}_k)] = 0,$$

for all $k \in V$ as claimed in equation (16). $\qquad\square$

## B Omitted Proofs and Materials: Section 4

In this section, we develop the necessary technical framework for the proof of Theorem 4.1. Theorem 4.1 is based upon two critical lemmas and properties of both KL divergence and rescaled zero-sum polymatrix games.

First, recall that in KL-divergence is not symmetric, i.e., it need not hold that $D_{\mathrm{KL}}(\mathbf{p} \parallel \mathbf{x}(t)) = D_{\mathrm{KL}}(\mathbf{x}(t) \parallel \mathbf{p})$. However, KL-divergence does obey the following property.

**Property 1.** *Let $k \in V$ and let $\mathbf{p}_k, \mathbf{x}_k$ be interior points of $\Delta_k$. Then, it holds that*

$$D_{KL}(\mathbf{p}_k \parallel \mathbf{x}_k) + D_{KL}(\mathbf{x}_k \parallel \mathbf{p}_k) = (\mathbf{x}_k - \mathbf{p}_k)^\top [\ln(\mathbf{x}_k) - \ln(\mathbf{p}_k)].$$

*Proof.* It is immediate to check that

$$
\begin{aligned}
(\mathbf{x}_k - \mathbf{p}_k)^\top [\ln(\mathbf{x}_k) - \ln(\mathbf{p}_k)] &= \mathbf{x}_k^\top \ln\left(\frac{\mathbf{x}_k}{\mathbf{p}_k}\right) - \mathbf{p}_k^\top \ln\left(\frac{\mathbf{x}_k}{\mathbf{p}_k}\right) \\
&= \mathbf{x}_k^\top \ln\left(\frac{\mathbf{x}_k}{\mathbf{p}_k}\right) + \mathbf{p}_k^\top \ln\left(\frac{\mathbf{p}_k}{\mathbf{x}_k}\right) \\
&= D_{\mathrm{KL}}(\mathbf{x}_k \parallel \mathbf{p}_k) + D_{\mathrm{KL}}(\mathbf{p}_k \parallel \mathbf{x}_k),
\end{aligned}
$$

as claimed. $\qquad\square$

**Lemma 4.2.** *Let $k \in V$. The time-derivative of the $D_{KL}$-divergence between the $k$-th component, $\mathbf{p}_k \in \Delta_k$, of a QRE $\mathbf{p} \in \Delta$ of $\Gamma$, and the $k$-th component, $\mathbf{x}_k(t) \in \Delta_k$ of a system trajectory with $\mathbf{x}(0)$ an interior point, is given by*

$$\frac{d}{dt} D_{KL}(\mathbf{p}_k \parallel \mathbf{x}_k(t)) = (\mathbf{x}_k - \mathbf{p}_k)^\top [r_k(\mathbf{x}_{-k}) - r_k(\mathbf{p}_{-k})] - T_k [D_{KL}(\mathbf{p}_k \parallel \mathbf{x}_k) + D_{KL}(\mathbf{x}_k \parallel \mathbf{p}_k)]. \tag{17}$$

*Proof.* The time derivative of the term $D_{\mathrm{KL}}(\mathbf{p}_k \parallel \mathbf{x}_k)$ for $k \in V$ can be calculated as follows (note that after the first line, we omit the dependence of $\mathbf{x}$ on $t$ to simplify notation)

$$
\begin{aligned}
\frac{d}{dt} D_{\mathrm{KL}}(\mathbf{p}_k \parallel \mathbf{x}_k(t)) &= -\sum_{i \in S_k} p_{ki} \frac{d}{dt}(\ln(x_{ki}(t))) = -\sum_{i \in S_k} p_{ki} \frac{\dot{x}_{ki}(t)}{x_{ki}(t)} \\
&= -\sum_{i \in S_k} p_{ki} \left[r_{ki}(\mathbf{x}_{-k}) - \mathbf{x}_k^\top r_k(\mathbf{x}_{-k}) + T_k\left(-\ln(x_{ki}) + \mathbf{x}_k^\top \ln(\mathbf{x}_k)\right)\right] \\
&= -\left[\mathbf{p}_k^\top r_k(\mathbf{x}_{-k}) - \mathbf{x}_k^\top r_k(\mathbf{x}_{-k}) + T_k\left(-\mathbf{p}_k^\top \ln(\mathbf{x}_k) + \mathbf{x}_k^\top \ln(\mathbf{x}_k)\right)\right] \\
&= (\mathbf{x}_k - \mathbf{p}_k)^\top r_k(\mathbf{x}_{-k}) - T_k(\mathbf{x}_k - \mathbf{p}_k)^\top \ln(\mathbf{x}_k) \\
&= (\mathbf{x}_k - \mathbf{p}_k)^\top [r_k(\mathbf{x}_{-k}) - T_k \ln(\mathbf{x}_k)].
\end{aligned}
$$

Using equation (16), i.e., that

$$(\mathbf{x}_k - \mathbf{p}_k)^\top [r_k(\mathbf{p}_{-k}) - T_k \ln(\mathbf{p}_k)] = 0,$$

we can write the time derivative of $D_{\mathrm{KL}}$ as follows

$$\frac{d}{dt} D_{\mathrm{KL}}(\mathbf{p}_k \parallel \mathbf{x}_k(t)) = (\mathbf{x}_k - \mathbf{p}_k)^\top [r_k(\mathbf{x}_{-k}) - T_k \ln(\mathbf{x}_k)] - (\mathbf{x}_k - \mathbf{p}_k)^\top [r_k(\mathbf{p}_{-k}) - T_k \ln(\mathbf{p}_k)]$$

$$= (\mathbf{x}_k - \mathbf{p}_k)^\top [r_k(\mathbf{x}_{-k}) - r_k(\mathbf{p}_{-k})] - T_k(\mathbf{x}_k - \mathbf{p}_k)^\top [\ln \mathbf{x}_k - \ln \mathbf{p}_k]$$

$$= (\mathbf{x}_k - \mathbf{p}_k)^\top [r_k(\mathbf{x}_{-k}) - r_k(\mathbf{p}_{-k})] - T_k [D_{\mathrm{KL}}(\mathbf{p}_k \parallel \mathbf{x}_k) + D_{\mathrm{KL}}(\mathbf{x}_k \parallel \mathbf{p}_k)]$$

where the last equation holds due to Property 1. This concludes the proof of the Lemma. $\qquad\square$

**Lemma 4.3.** *Let* $\mathbf{p} = (\mathbf{p}_k)_{k \in V}$ *be a QRE of* $\Gamma$ *and let* $\mathbf{x} = (\mathbf{x}_k)_{k \in V}$*. Then, it holds that*

$$\sum_{k \in V} w_k \left[ \mathbf{x}_k^\top r_k(\mathbf{p}_{-k}) + \mathbf{p}_k^\top r_k(\mathbf{x}_{-k}) \right] = 0. \tag{18}$$

Before proceeding with the proof of Lemma 4.3, note that if there only two agents, i.e., if $V = \{1, 2\}$, then $\mathbf{p} = (\mathbf{p}_1, \mathbf{p}_2)$ and $\mathbf{x} = (\mathbf{x}_1, \mathbf{x}_2)$, and it is rather immediate to check the validity equation (18), since

$$\sum_{k=1,2} w_k [\mathbf{x}_k^\top r_k(\mathbf{p}_{-k}) + \mathbf{p}_k^\top r_k(\mathbf{x}_{-k})] =$$

$$= w_1 \left[ \mathbf{x}_1^\top r_1(\mathbf{p}_2) + \mathbf{p}_1^\top r_1(\mathbf{x}_2) \right] + w_2 \left[ \mathbf{x}_2^\top r_2(\mathbf{p}_1) + w_2 \mathbf{p}_2^\top r_2(\mathbf{x}_1) \right]$$

$$= \left[ w_1 \mathbf{x}_1^\top r_1(\mathbf{p}_2) + w_2 \mathbf{p}_2^\top r_2(\mathbf{x}_1) \right] + \left[ w_2 \mathbf{x}_2^\top r_2(\mathbf{p}_1) + w_2 \mathbf{p}_1^\top r_1(\mathbf{x}_2) \right]$$

$$= \underbrace{\sum_{k=1,2} w_k u_k(\mathbf{x}_1, \mathbf{p}_2)}_{= 0 \text{ by equation (2)}} + \underbrace{\sum_{k=1,2} w_k u_k(\mathbf{p}_1, \mathbf{x}_2)}_{= 0 \text{ by equation (2)}} = 0.$$

In particular, the last equation holds because the summations are over all payoffs in the strategy profiles $(\mathbf{x}_1, \mathbf{p}_2)$ and $(\mathbf{p}_1, \mathbf{x}_2)$. When we have more than two agents, this argument does not hold since the summation is over different strategy profiles. However, it still holds that the summation is equal to zero. To show this, we will need to apply the following property that has been established in [12] for zero-sum polymatrix games to the case of weighted zero-sum polymatrix games (the extension is rather straightforward as we show below). To state the property in the general case, we will use the following definition.

**Definition 2** ($w_k$-Payoff equivalence)**.** Consider two arbitrary games $\Gamma = ((V, E), (S_k, u_k)_{k \in V})$ and $\Gamma' = ((V, E), (S_k, u'_k)_{k \in V})$. We will say that $\Gamma$ is $w_k$-payoff equivalent to $\Gamma'$ if there exist positive constants $w_k, k \in V$ so that

$$u_k(\mathbf{x}) = w_k u'_k(\mathbf{x}), \quad \text{for all } x \in \Delta.$$

**Property 2** (Payoff equivalent transformation [12].)**.** *Let* $\Gamma = \left( (V, E), (S_k, w_k)_{k \in V}, (\mathbf{A}_{kl})_{[k,l] \in E} \right)$ *be a rescaled zero-sum polymatrix game. Then,* $\Gamma$ *is* $1/w_k$*-payoff equivalent to a pairwise constant-sum (unweighted) polymatrix game* $\hat{\Gamma} = \left( (V, E), (S_k)_{k \in V}, \left( \hat{\mathbf{A}}_{kl} \right)_{[k,l] \in E} \right)$*, i.e., a game in which every two-agent game* $[k, l] \in E$ *is constant-sum and all these constants sum up to zero. Specifically, for all* $k, l \in V$ *with* $[k, l] \in E$*, there exist payoff matrices* $\hat{\mathbf{A}}_{kl} = (\hat{a}_{kl}(s_k, s_l))_{\{s_k \in S_k, s_l \in S_l\}}$ *and constants* $c_{kl} \in \mathbb{R}$*, so that*

$$\hat{a}_{kl}(s_k, s_l) + \hat{a}_{lk}(s_l, s_k) = c_{kl}, \quad \text{for all } s_k \in S_k, s_l \in S_l, \tag{19}$$

*with*

$$\sum_{[k,l] \in E} c_{kl} = 0, \tag{20}$$

*and*

$$\hat{r}_{ki}(\mathbf{x}_{-k}) := \sum_{[k,l] \in E} \{\hat{\mathbf{A}}_{kl} \mathbf{x}_l\}_i = \sum_{[k,l] \in E} w_k \{\mathbf{A}_{kl} \mathbf{x}_l\}_i = w_k r_{ki}(\mathbf{x}_{-k}), \tag{21}$$

*for all* $k \in V$ *and all* $i \in S_k$*, i.e., for every pure (and hence, also for every mixed) strategy profile, the payoff of each agent* $k \in V$ *in* $\hat{\Gamma}$ *is equal to* $1/w_k$ *their payoff in the original game* $\Gamma$*.*

*Proof.* As mentioned above, all claims of Property 2 have been established for (unweighted) zero-sum polymatrix games in [12]. Thus, it remains to show that the proof extends to the weighted case. To see this, consider a weighted zero-sum polymatrix game $\Gamma$ with weights $\mathbf{w} = (w_k)_{k \in V}$, payoff matrices $(\mathbf{A}_{kl})_{[k,l] \in E}$ and utilities $u_k, k \in V$ as in equation (1) and define the transformed game $\Gamma^{\mathbf{B}}$ with payoff matrices $(\mathbf{B}_{kl})_{[k,l] \in E}$ given by

$$\mathbf{B}_{kl} := w_k \mathbf{A}_{kl}, \quad \text{for all } k \in V, [k,l] \in E,$$

and utilities $u_k^{\mathbf{B}}, k \in V$. Then, it follows from equation (1) that $u_k(\mathbf{x}) = \frac{1}{w_k} u_k^{\mathbf{B}}(\mathbf{x})$ for all $\mathbf{x} \in \Delta$ and $k \in V$ since

$$u_k^{\mathbf{B}}(\mathbf{x}) = \sum_{[k,l] \in E} \mathbf{x}_k^\top \mathbf{B}_{kl} \mathbf{x}_l = \sum_{[k,l] \in E} \mathbf{x}_k^\top (w_k \mathbf{A}_{kl}) \mathbf{x}_l = w_k \sum_{[k,l] \in E} \mathbf{x}_k^\top \mathbf{A}_{kl} \mathbf{x}_l = w_k u_k(\mathbf{x}).$$

Moreover, $\Gamma^{\mathbf{B}}$ is an *unweighted* zero-sum game since

$$\sum_{k \in V} u_k^{\mathbf{B}}(\mathbf{x}) = \sum_{k \in V} w_k u_k(\mathbf{x}) = 0,$$

where the last equality follows from the fact that $\Gamma$ is a weighted zero-sum polymatrix game with weights $(w_k)_{k \in V}$. Thus, transforming $\Gamma$ to $\Gamma^{\mathbf{B}}$ and then applying the transformation of [12] on $\Gamma^{\mathbf{B}}$ to obtain the payoff equivalent (to $\Gamma^{\mathbf{B}}$) game $\hat{\Gamma}$ with payoff matrices $\hat{\mathbf{A}}_{kl}, [k,l] \in E$ yields the result, i.e., the $1/w_k$-payoff equivalence between the original weighted zero-sum polymatrix game $\Gamma$ and the pairwise constant-sum, unweighted polymatrix game $\hat{\Gamma}$. $\square$

Using Property 2, we can now prove Lemma 4.3 for general $k \geq 2$.

*Proof of Lemma 4.3.* Equation (21) in Property 2 implies that

$$\sum_{k \in V} w_k \left[ \mathbf{x}_k^\top r_k (\mathbf{p}_{-k}) + \mathbf{p}_k^\top r_k (\mathbf{x}_{-k}) \right] \overset{(21)}{=} \sum_{k \in V} w_k \left[ \mathbf{x}_k^\top \left( \frac{1}{w_k} \hat{r}_k(\mathbf{x}_{-p}) \right) + \mathbf{p}_k^\top \left( \frac{1}{w_k} \hat{r}_k(\mathbf{x}_{-k}) \right) \right]$$

$$= \sum_{k \in V} \sum_{[k,l] \in E} \left[ \mathbf{x}_k^\top \hat{\mathbf{A}}_{kl} \mathbf{p}_l + \mathbf{p}_k^\top \hat{\mathbf{A}}_{kl} \mathbf{x}_l \right]$$

$$= \sum_{[k,l] \in E} \left[ \mathbf{x}_k^\top \hat{\mathbf{A}}_{kl} \mathbf{p}_l + \mathbf{p}_k^\top \hat{\mathbf{A}}_{kl} \mathbf{x}_l + \mathbf{x}_l^\top \hat{\mathbf{A}}_{lk} \mathbf{p}_k + \mathbf{p}_l^\top \hat{\mathbf{A}}_{kl} \mathbf{x}_k \right]$$

$$= \sum_{[k,l] \in E} \underbrace{\left[ \mathbf{x}_k^\top \hat{\mathbf{A}}_{kl} \mathbf{p}_l + \mathbf{p}_l^\top \hat{\mathbf{A}}_{kl} \mathbf{x}_k \right]}_{= c_{kl} \text{ by equation (19)}} + \underbrace{\left[ \mathbf{x}_l^\top \hat{\mathbf{A}}_{lk} \mathbf{p}_k + \mathbf{p}_k^\top \hat{\mathbf{A}}_{kl} \mathbf{x}_l \right]}_{= c_{kl} \text{ by equation (19)}}$$

$$= 2 \sum_{[k,l] \in E} c_{kl} = 0,$$

where the last equality holds by equation (20). $\square$

## B.1 Proof of Theorem 4.1

Combining Lemmas 4.2 and 4.3, we can now restate and prove Theorem 4.1.

**Theorem 4.1.** *Let $\Gamma$ be a rescaled zero-sum polymatrix game, with positive exploration rates $T_k$. There exists a unique QRE $\mathbf{p}$ such that if $\mathbf{x}(t)$ is any trajectory of the associated Q-learning dynamics $\dot{\mathbf{x}} = f(\mathbf{x})$, with $f_i$ is given via* (QLD)*, where $\mathbf{x}(0)$ is an interior point, then $\mathbf{x}(t)$ converges to $\mathbf{p}$ exponentially fast. In particular, we have that*

$$\frac{d}{dt} D_{KL(w)}(\mathbf{p} \parallel \mathbf{x}(t)) = - \sum_{k \in V} w_k T_k \left[ D_{KL}(\mathbf{p}_k \parallel \mathbf{x}_k) + D_{KL}(\mathbf{x}_k \parallel \mathbf{p}_k) \right]. \tag{22}$$

*Proof.* Proposition 3.2 states that there must exist some a QRE equilibrium $\mathbf{p} = (\mathbf{p}_k)$. We first establish (22), from which the remaining statements will follow. The time derivative of the KL-divergence between $\mathbf{p}$ and $\mathbf{x}(t)$ is given by (as in the proof of Lemma 4.2, we again omit the

dependence of $\mathbf{x}$ on $t$ after the first line to simplify notation)

$$\frac{d}{dt}D_{\mathrm{KL}}(\mathbf{p} \parallel \mathbf{x}(t)) = \sum_{k \in V} \frac{d}{dt} w_k D_{\mathrm{KL}}(\mathbf{p}_k \parallel \mathbf{x}_k)$$

$$= \sum_{k \in V} w_k \left[ (\mathbf{x}_k - \mathbf{p}_k)^\top \left[ r_k(\mathbf{x}_{-k}) - r_k(\mathbf{p}_{-k}) \right] - T_k \left( D_{\mathrm{KL}}(\mathbf{p}_k \parallel \mathbf{x}_k) + D_{\mathrm{KL}}(\mathbf{x}_k \parallel \mathbf{p}_k) \right) \right]$$

$$= \sum_{k \in V} w_k (\mathbf{x}_k - \mathbf{p}_k)^\top \left[ r_k(\mathbf{x}_{-k}) - r_k(\mathbf{p}_{-k}) \right] - \sum_{k \in V} w_k T_k \left[ D_{\mathrm{KL}}(\mathbf{p}_k \parallel \mathbf{x}_k) + D_{\mathrm{KL}}(\mathbf{x}_k \parallel \mathbf{p}_k) \right].$$

The first term in the right-hand side of the last equation is equal to zero, since

$$\sum_{k \in V} w_k (\mathbf{x}_k - \mathbf{p}_k)^\top \left[ r_k(\mathbf{x}_{-k}) - r_k(\mathbf{p}_{-k}) \right] = \sum_{k \in V} w_k \mathbf{x}_k^\top r_k(\mathbf{x}_{-k}) + \sum_{k \in V} w_k \mathbf{p}_k^\top r_k(\mathbf{p}_{-k})$$

$$+ \sum_{k \in V} w_k \left[ \mathbf{x}_k^\top r_k(\mathbf{p}_{-k}) + \mathbf{p}_k^\top r_k(\mathbf{x}_{-k}) \right] = 0,$$

where in the last equality, we used the zero-sum property (cf. equation (2)) to conclude that the terms $\sum_{k \in V} w_k \mathbf{x}_k^\top r_k(\mathbf{x}_{-k})$ and $\sum_{k \in V} w_k \mathbf{p}_k^\top r_k(\mathbf{p}_{-k})$ are equal to 0 and equation (18) in Lemma 4.3 to conclude that the term $\sum_{k \in V} w_k \left[ \mathbf{x}_k^\top r_k(\mathbf{p}_{-k}) + \mathbf{p}_k^\top r_k(\mathbf{x}_{-k}) \right]$ is also equal to 0. Thus,

$$\frac{d}{dt}D_{\mathrm{KL}}(\mathbf{p} \parallel \mathbf{x}(t)) = - \sum_{k \in V} w_k T_k \left[ D_{\mathrm{KL}}(\mathbf{p}_k \parallel \mathbf{x}_k) + D_{\mathrm{KL}}(\mathbf{x}_k \parallel \mathbf{p}_k) \right], \tag{23}$$

which implies that $\frac{d}{dt}D_{\mathrm{KL}}(\mathbf{p} \parallel \mathbf{x}) < 0$ for all $\mathbf{x} \neq \mathbf{p}$ as $T_k > 0$ for each $k$. In other words, $\Phi(\mathbf{x}) := D_{\mathrm{KL}}(\mathbf{p} \parallel \mathbf{x})$ obeys the properties i) $\Phi(\mathbf{p}) = 0$, ii) $\Phi(\mathbf{x}) > 0$ if $\mathbf{x} \neq \mathbf{p}$ and iii) $\dot{\Phi}(\mathbf{x}) < 0$ for $\mathbf{x} \neq \mathbf{p}$, i.e., $\Phi$ is a Lyapunov function for the Q-learning dynamics (QLD). In particular, since $D_{\mathrm{KL}}(\mathbf{x}_k \parallel \mathbf{p}_k) > 0$ as long as the distributions $\mathbf{p}_k, \mathbf{x}_k$ are not equal, then

$$\dot{\Phi}(x) \leq - \min_k T_k \cdot \Phi(x), \tag{24}$$

thus, the KL-divergence converges to zero exponentially fast. As any QRE $\mathbf{p}' \neq \mathbf{p}$ must also satisfy $\dot{\Phi}(\mathbf{p}') = \nabla \Phi \cdot f(\mathbf{p}') = 0$ (as $\mathbf{p}'$ is a fixed point for $\dot{\mathbf{x}} = f(\mathbf{x})$, where $f$ is given in (QLD)), but we have that $\dot{\Phi}(\mathbf{x}) < 0$ for all $\mathbf{x} \neq \mathbf{p}$, it follows that $\mathbf{p}$ is unique. $\qquad\square$

## C  Additional Experiments

In the section, we present additional simulations and calculations that complement our experimental results in Section 5 in the main paper.

### C.1  Q-learning dynamics in the AMPs game

We start with Figure 6 which completes the possible combinations of the two representative policies that we consider, CLR-1 and ETE, in the AMPs game (cf. Section 5). The plots are similar to that in Figure 1.

### C.2  Q-learning dynamics in two-agent games with more actions

We next turn to visualizations of the Q-learning in games with two-agents but more than two actions for each agent. For this purpose, we consider the (symmetric) zero-sum game, Rock-Paper-Scissors (RPS) with payoff matrices given by

$$\mathbf{A} = \begin{pmatrix} 0 & -1 & 1 \\ 1 & 0 & -1 \\ -1 & 1 & 0 \end{pmatrix}, \quad \mathbf{B} = -\mathbf{A}^\top. \tag{RPS}$$

The RPS game has a unique (interior) Nash equilibrium given by $(p^*, q^*) = ((1/3, 1/3, 1/3), (1/3, 1/3, 1/3))$. This is also the unique QRE for any positive exploration rates (since exploration

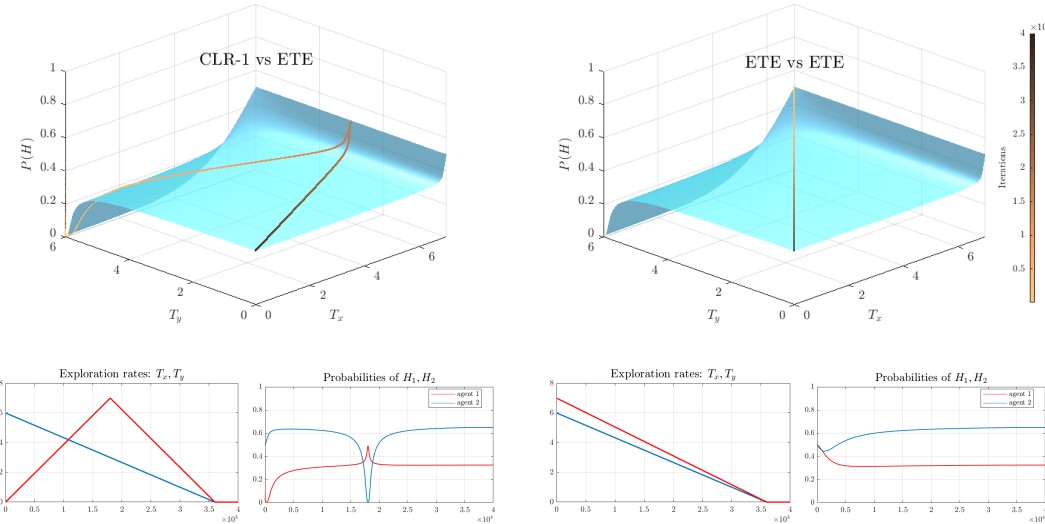

Figure 6: QRE surface and exploration paths (upper panels) for two different exploration policy profiles (lower panels) in the Asymmetric MPs game. As in Figure 1 in the main part, the sequence of play converges to the unique QRE for any combination of exploration policies (CLR-1 and ETE). As the exploration rates decrease zero, the sequence of play converges to the unique Nash equilibrium of the game.

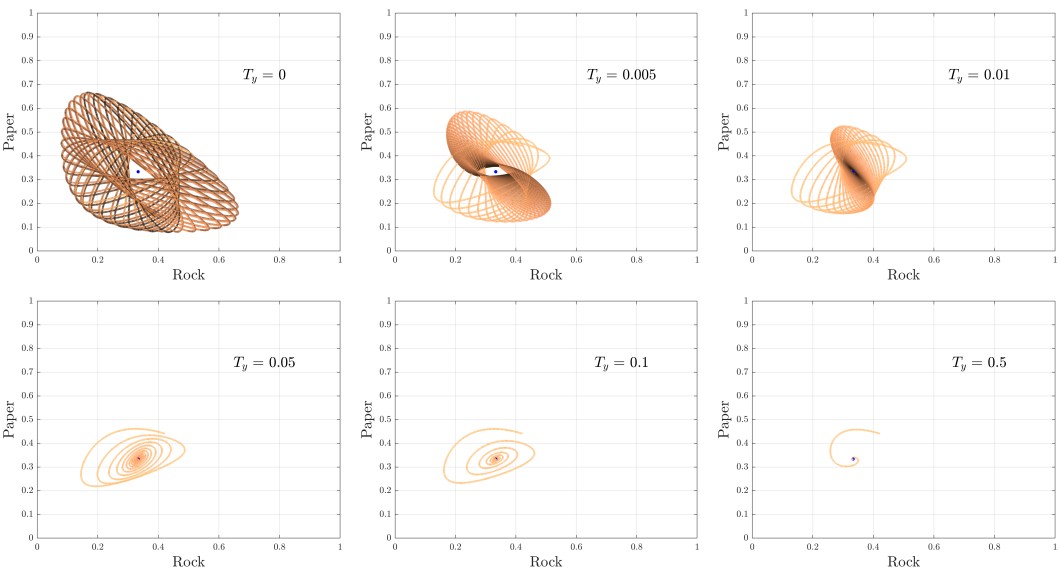

Figure 7: Q-learning dynamics in Rock-Paper-Scissors for $T_x = 0$ (no exploration by $x$-agent) and six different exploration rates, $T_y \geq 0$ by $y$-agent. The light to dark trajectories (with darkening color indicating increasing time) show the choice distribution for the $x$-agent in the Rock-Paper space.

favors the uniform distribution). In Figure 7, we visualize the Q-learning dynamics (cf. equation (QLD)) for $T_x = 0$ and various values of $T_y$. The setup and the plots are similar to that of Figure 2.

The trajectories in Figure 7 have been generated for random initial conditions (similar plots are obtained for any other initial condition) and show the choice distribution of the $x$-agent at each time point of the simulation (we used $2 \times 10^7$ iterations with a step 0.0003 for the discretization of the continuous time ODE in equation (QLD)). We obtain similar plots for the exploring agent ($y$-agent), see Figure 8. When $T_y = 0$, the Q-learning dynamics reduce to the replicator dynamics, and we

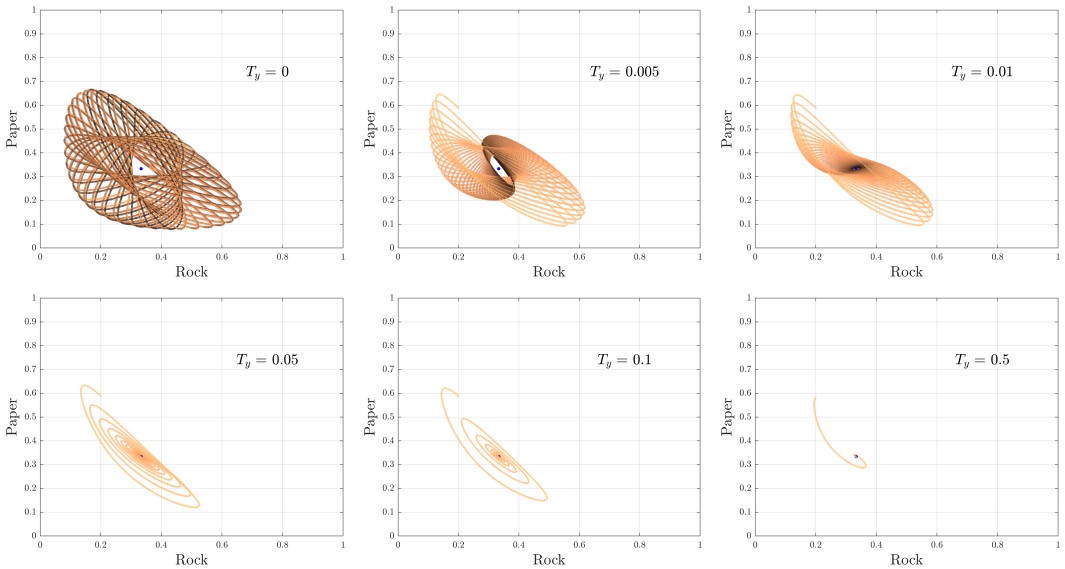

Figure 8: Q-learning trajectories for the $y$-agent in the instances of the RPS game that are shown in Figure 7.

recover their cyclic behavior (Poincaré Recurrence) around the unique interior Nash equilibrium (upper left panel) (see [43] and references therein). In all other cases, exploration by one agent suffices for the convergence of the joint-learning dynamics as in the AMPs game (in panels 2 and 3, the dynamics spiral inwards and will eventually converge to the QRE (blue dot)). As we saw in Section 5 this is in sharp contrast to the (MMG) game in which exploration (even) by several agents was not sufficient for the convergence of Q-learning to a single QRE. It is worth mentioning that this behavior of the Q-learning dynamics in RPS does not rely on the symmetry of the game. We obtain similar plots for the modified RPS game (with asymmetric Nash equilibrium) (not depicted here).

### C.3 Edge Case: Exploration by one agent in $2 \times 2$ games

Up to now, we have treated the case of exploration by only agent experimentally (cf. Figures 2, 7 and 8). In this part, we consider the edge-case in which only one of two agents is exploring in $2x2$ games which is also analytically tractable. Our result is presented in Proposition C.1 and the ensuing intuition is summarized in Remark 1. For the following (technical) calculations, we will use the notation

$$\mathbf{A} = \begin{pmatrix} a_{11} & a_{12} \\ a_{21} & a_{22} \end{pmatrix}, \quad \mathbf{B} = \begin{pmatrix} b_{11} & b_{12} \\ b_{21} & b_{22} \end{pmatrix}.$$

We assume that the game has a (unique) interior Nash equilibrium. This implies (without loss of generality, see e.g., [42]) that

$$a_{11} > a_{21},\ a_{12} < a_{22}, \quad \text{and} \quad b_{11} < b_{21},\ b_{12} > b_{22}. \tag{25}$$

By letting $a_1 := a_{12} - a_{22}$ and $a_2 := a_{12} + a_{21} - a_{11} - a_{22}$ with $a_1, a_2 < 0$ and $b_1 := b_{12} - b_{22}$ and $b_2 := b_{12} + b_{21} - b_{11} - b_{22}$ with $b_1, b_2 > 0$, we have that the unique Nash equilibrium, $(\mathbf{p}, \mathbf{q})$, with $\mathbf{p} = (p, 1-p), \mathbf{q} = (q, 1-q)$ of this game is given by $q = \frac{a_1}{a_2}$ and $p = \frac{b_1}{b_2}$ with $p, q \in (0, 1)$ by assumption.

Since agents have two strategies, we can represent their mixed strategies by the vectors $\mathbf{x} = (x, 1-x)$ and $\mathbf{y} = (y, 1-y)$ with $x, y \in [0, 1]$. Using this notation and under the assumption that $T_x = 0$, i.e., that the $x$ agent is not exploring, the dynamics for the $x$ agent in equation (QLD) take the form

$$\dot{x} = x \left[ (1,\ 0)\mathbf{A} \begin{pmatrix} y \\ 1-y \end{pmatrix} - (x,\ 1-x)\mathbf{A} \begin{pmatrix} y \\ 1-y \end{pmatrix} \right] = x(1-x) \left[ a_1 - a_2 y \right]$$

by assumption (25). Similarly, the dynamics for the $y$ agent in equation (QLD) take the form

$$\dot{y} = y\left[(1,\ 0)\mathbf{B}\begin{pmatrix} x \\ 1-x \end{pmatrix} - (y,\ 1-y)\mathbf{B}\begin{pmatrix} x \\ 1-x \end{pmatrix} + T_y(y \ln y + (1-y) \ln 1 - y - \ln y)\right]$$

$$= y(1-y)\left[b_1 - b_2 x + T_y \ln\left(\frac{1}{y} - 1\right)\right]$$

where we defined $b_1 := b_{12} - b_{22}$ and $b_2 := b_{12} + b_{21} - b_{11} - b_{22}$ with $b_1, b_2 > 0$ by assumption (25). For the $y$ agent, we will consider various values for $T_y \geq 0$. Putting these together, we obtain the system

$$\dot{x} = x(1-x)\left[a_1 - a_2 y\right]$$
$$\dot{y} = y(1-y)\left[b_1 - b_2 x + T_y \ln\left(\frac{1}{y} - 1\right)\right] \tag{26}$$

with $x, y \in [0, 1]$ and $T_y \geq 0$. It is immediate that all four points $(x, y) \in \{0, 1\} \times \{0, 1\}$ are resting point of the system. All these points lie on the boundary. For the system to have an interior resting point, we have the conditions

$$y = \frac{a_1}{a_2}, \quad \text{and} \quad x = \frac{1}{b_2}\left[b_1 + T_y \ln\left(\frac{a_2}{a_1} - 1\right)\right]. \tag{27}$$

The condition for $y$ is always satisfied by assumption (25). In particular, this yields the Nash equilibrium strategy for the $y$ agent. However, the condition for $x$ may yield an $x$ that does not lie within $(0, 1)$. For $T_y = 0$, the condition becomes $x = b_1/b_2$ which is the Nash equilibrium strategy for the $x$ agent. By assumption, this is strictly between 0 and 1. However, $x$ depends linearly on $T_y$ and depending on whether $a_2/a_1 > 2$ or $a_2/a_1 < 2$ it either increases or decreases in $T_y$.[2] Thus, there exists a critical threshold, $T_y^*$, for which $x$ hits the boundary of $[0, 1]$, i.e., it either becomes 0 or 1. Assume that $x = 1$ without loss of generality. At that point, the upper equation in (26) is satisfied regardless of whether $y = a_1/a_2$. Thus, we turn to the lower equation in (26) to obtain a condition for $y$, which yields

$$b_1 - b_2 \cdot 1 + T_y \ln\left(\frac{1}{y} - 1\right) = 0 \iff y = \left[1 + \exp\left((b_2 - b_1)/T_y\right)\right]^{-1}.$$

Note, that in all the above cases, if an interior resting point exists (for either or both agents), then it is unique. We summarize our findings for the $2 \times 2$ case in Proposition C.1.

**Proposition C.1.** *Let $\Gamma = (\{1, 2\}, \mathbf{A}, \mathbf{B})$ with $\mathbf{A}, \mathbf{B} \in \mathbb{R}^{2 \times 2}$ be a two-agent, two-strategy $(2 \times 2)$ game with $a_2 > 2a_1$ and a unique interior Nash equilibrium, i.e., such that condition (25) holds, and let*

$$T_y^{crit} := (b_2 - b_1) \cdot \left[\ln\left(\frac{a_2}{a_1} - 1\right)\right]^{-1}.$$

*If $T_x = 0$, then for any interior starting point $(x_0, y_0) \in (0, 1)$, the fixed points $(\mathbf{p}, \mathbf{q}) = ((p, 1 - p), (q, 1 - q))$ with $p, q \in [0, 1]$ of the Q-learning dynamics*

$$\dot{x} = x(1-x)\left[a_1 - a_2 y\right]$$
$$\dot{y} = y(1-y)\left[b_1 - b_2 x + T_y \ln\left(\frac{1}{y} - 1\right)\right]$$

*depend on $T_y$ as follows:*

- *if $T_y = 0$, then the dynamics are cyclic, i.e., they do not have a resting point.*
- *if $0 < T_y < T_y^{crit}$, then they have a unique interior resting point which is given by*

$$(p, q) = \left(\frac{1}{b_2}\left[b_1 + T_y \ln\left(\frac{a_2}{a_1} - 1\right)\right],\ \frac{a_1}{a_2}\right) \quad \text{with } p, q \in (0, 1).$$

---

[2]The case $a_2 = 2a_1$ is trivial since it implies that $y = 1/2$ and $x = b_1/b_2$ regardless of $T_y$.

- *otherwise, i.e., if $T_y > T_y^{crit}$, then they have a resting point that lies in the interior only for the exploring agent which is given by*

$$(p, q) = \left(1, \ [1 + \exp\left((b_2 - b_1)/T_y\right)]^{-1}\right), \quad \text{with } q \in (0, 1).$$

*In particular, as $T_y \to \infty$, it holds that $q \to 1/2$.*

If $a_2 < 2a_1$, then $x$ is decreasing in $T_y$ and hence, at $T_y^{crit}$ it becomes $0$ (rather than $1$). In this case, solving equation (27) for $T_y^{crit}$ yields $T_y^{crit} = -b_1 \cdot \left[\ln\left(\frac{a_2}{a_1} - 1\right)\right]^{-1}$. By renaming the strategies of the $x$ agent, this case is equivalent to the one presented in Proposition C.1.

*Remark* 1 (Intuition of Proposition C.1). The main takeaway of Proposition C.1 is the qualitative description of the resting points of the Q-learning dynamics for any interior starting point. When $T_y = 0$, the dynamics are the replicator dynamics, which are well-known to cycle around the unique interior Nash equilibrium in this case [7]. If $T_y > 0$, then this suffices to ensure convergence to the unique interior QRE (cf. Theorem 4.1). As long as $T_y$ is small enough, i.e., smaller than a critical value, $T_y^{\text{crit}}$, the $y$ component of the QRE corresponds precisely to the Nash equilibrium (mixed) strategy for the $y$-agent (the exploring agent) and an interior value for the $x$-agent (different that her Nash equilibrium mixture). This value is increasing (assuming that $a_2 > 2a_1$, otherwise it is decreasing or constant if $a_2 = 2a_1$) in the exploration rate $T_y$ of the $y$-agent. This implies, that the QRE component for the $x$-agent approaches the boundary. After exploration by the $y$-agent exceeds the critical threshold $T_y^{\text{crit}}$, the $x$-agent starts playing a pure strategy. At that point onward (i.e., for larger exploration rates), the utility of the $y$ agent is dominated by her exploration term and her mixture (at QRE) starts to approach the uniform distribution.

The result of Proposition C.1 is illustrated in the main part of the paper via Asymmetric Matching Pennies (AMPs) game in Figure 2.

**Exploration by one agent in games with more than** $2$ **strategies.** The general case of more than $2$ strategies for each agent can be qualitatively different to the $2 \times 2$ case presented above. To see this, consider a variation of Rock-Paper-Scissors (RPS) in which the second agent has a copy of strategy *scissors* with exactly the same utility as the original strategy *scissors*. The modified game has infinite many Nash equilibria in which the first agent plays $p^* = (1/3, 1/3, 1/3)$ as in the original game and the second agent plays $q^* = (1/3, 1/3, x, 1/3 - x)$ for any $x \in [0, 1/3]$. Now assume that the first agent has a positive exploration rate and that the second agent has zero exploration rate. As in the unmodified game, the uniform distribution is still optimal for the first agent since it also maximizes the entropy but now any $q^* = (1/3, 1/3, x, 1/3 - x)$ for $x \in [0, 1/3]$, remains an equilibrium strategy for the second agent.

### C.4 3D Visualization of the Lyapunov function in $n$-agent network games

To visualize the Lyapunov function in zero-sum network games with $n$-agents with strictly positive exploration profiles, i.e., $T_k > 0$ for all $k \in V$, (KL-divergence from the current action profile to the unique QRE for that exploration profile), we adapt the dimension reduction method of [35] (cf. [34]). This yields panel 4 in Figure 4.

For an $n$-agent network game, with a fixed and strictly positive exploration profile, we start by determining its unique QRE, $q$. By our main result, Theorem 4.1, this can be done by simulating the Q-learning dynamics. Then, we select two vectors $u, v$ with $n$ random entries each in $(0, 1)$. The random entries of the vectors $u, v$ correspond to the probability with which each agent selects their first action (here $H$). Instead of forming convex combinations of these random vectors and plotting the Lyapunov function across this (randomly selected) space, we perform the following transformation that allows for a more comprehensive snapshot of the whole joint action space. Specifically, we map each coordinate $u(k)$ (and similarly for $v$) with $k \in V$ to $\tilde{u}(k) = \ln u(k)/(1 - u(k))$. Then, we form linear combinations of the transformed vectors $\tilde{u}, \tilde{v}$ using real-valued scalars, $\alpha, \beta \in \mathbb{R}$. Note that the all-zero vector (in the transformed space) corresponds to the uniform distribution for each agent (in the choice distribution space). Finally, we map the resulting point $z := \alpha \cdot \tilde{u} + \beta \cdot \tilde{v}$ from the transformed space back to the product simplex via the (coordinate-wise) transformation $z(k) \to \exp\left(z(k)\right)/(\exp\left(z(k)\right) + 1)$ and plot the potential at the resulting point (KL-divergence between that point and the unique QRE, $q$). We repeat the process for a range of both positive and

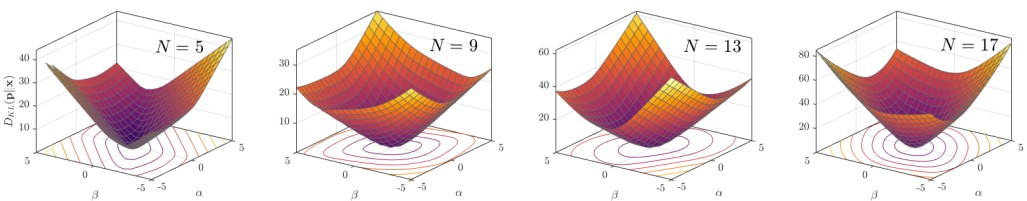

Figure 9: Snapshots of the Lyapunov function (KL-divergence between each choice distribution profile and the unique QRE) in four instances of the (MMG) game with fixed (and strictly positive) exploration profiles and different numbers of agents. In all cases, the Lyapunov (potential) function is convex with a unique minimizer at 0.

negative values for $\alpha$'s and $\beta$'s. This yields the $x - y$ coordinates in panel 4 and the evaluation yields the depicted 3D surface. The process is summarized in Algorithm 1.

---

**Algorithm 1** 3D Visualization of the Lyapunov function (KL-divergence)

---

**Input (network game):** number of agents, payoff matrices, (strictly positive) exploration rates.
**Output:** Snapshot of the Lyapunov function (KL-divergence).

1: **procedure** COMPUTE QRE($T_k, k = 1, \ldots, n$)
2:     $q \leftarrow$ unique QRE (e.g., by running (QLD))
3: $u, v$ generate random vectors with $n$ entries in $(0, 1)$.
4: **procedure** TRANSFORM VARIABLES($u, v, \alpha, \beta$)
5:     **for** $k \leftarrow V$ **do**
6:         $u(k) \leftarrow \ln\left(u(k)/(1 - u(k))\right)$
7:         $v(k) \leftarrow \ln\left(v(k)/(1 - v(k))\right)$
8:     $z \leftarrow \alpha \cdot u + \beta \cdot v$
9:     **for** $k \leftarrow V$ **do**
10:         $z(k) \leftarrow \exp\left(z(k)/(z(k) + 1)\right)$
11: **procedure** EVALUATE LYAPUNOV FUNCTION($q, z$)
12:     $D_{\text{KL}} = 0$
13:     **for** $k \in V$ **do**
14:         $D_{\text{KL}} \leftarrow D_{\text{KL}} + q(k) \ln \frac{q(k)}{z(k)} + (1 - q(k)) \ln \frac{1 - q(k)}{1 - z(k)}$
15:     **return** tuple $(\alpha, \beta, D_{\text{KL}})$
16: **plot** $\leftarrow (\alpha, \beta, D_{\text{KL}})$

---

A restriction of this method in $n$-agent games is that, for each point that it generates, it uses the same $\alpha, \beta$ to scale the transformed variables of *all* agents. Figure 9 shows snapshots of the Lyapunov function in four instances of the (MMG) game with fixed (and strictly positive) exploration profiles for different numbers of agents.

### C.5 Equilibrium selection in the network game

One question that is hard to tackle theoretically concerns the equilibrium selection as exploration rates converge to 0. As we saw in Theorem 4.1, when all agents have positive exploration rates, then there is a unique QRE and the joint-learning dynamics converge to that QRE. However, as exploration rates approach zero (for instance, after the exploration phase ends for all agents), it is not clear which equilibrium will be selected in the original game (as the limit of the sequence of the unique QRE for the different strictly positive exploration profiles).

In this part, we test this question experimentally in the (MMG) game of Section 5. We consider an instance with 3 non-dummy agents and different exploration policies for the agents. Recall that in this case, the original network game (with no exploration) has multiple Nash equilibria of the following form: the odd agents ($p_1$ and $p_3$) select $T$ with probability 1 and the even agent ($p_2$) plays an arbitrary strategy in $(0, 1)$ (i.e., probabilities of playing $H$). In any equilibrium, the payoff of $p_2$ is 0 whereas the payoffs of $p_1$ and $p_3$ sum up to 2. However, the crucial point is that the split of 2 between $p_1$ and

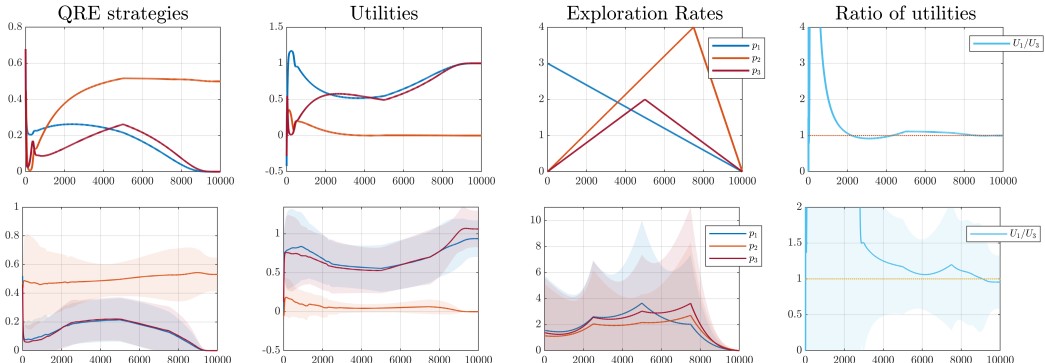

Figure 10: Effects of exploration on equilibrium selection in the (MMG) of Section 5. The upper panels show an individual run, and the bottom panels show averages (means and 1 standard deviation as shaded region around the mean) over 50 runs. Panels 1 to 3 show the probability of playing $H$ at QRE, the utilities and the exploration rates of the agents, respectively. The effect of exploration is shown via the ratio of utilities of $p_1$ and $p_3$ in the fourth panel of each row. Exploration by the even agent ($p_2$) leads that agent to select the $0.5$ strategy at equilibrium (when exploration drops back to $0$ by all agents) which results in a fair split (close to 1, dotted red line in panels 4) of the payoffs between agents $p_1$ and $p_3$.

$p_3$ critically depends on the strategy of $p_2$. In particular, we saw in Figure 5 that without exploration, the Q-learning dynamics can converge to any of these multiple equilibria, thus inducing arbitrary asymmetries between the payoffs of $p_1$ and $p_3$.

The results of one representative exploration scenario (with linearly changing exploration rates) and averages over 50 runs with randomly matched CLR-1 and ETE exploration policies are presented in Figure 10.

The main takeaway of these experiments is captured by the last panel "Ratio of utilities" of each row. Namely, sufficient exploration by the even agent (the agent with multiple equilibrium strategies) leads that agent to select a strategy close to the uniform one (here $0.5$ since there are two actions). In turn, this leads to a fair split of the stake between the odd numbered agents ($p_1$ and $p_3$). This is in sharp contrast to the case without exploration (cf. Figure 5 in the main part) in which any equilibrium (i.e., any strategy between $0, 1$) is a potential limit point of the dynamics for the even agent (thus, leading to arbitrary splits of the share between the odd numbered agents).

While these results cannot lead to a formal argument about the effect of exploration in equilibrium selection (when exploration goes back to zero for all agents), they highlight the importance of further studying equilibrium selection in competitive environments both experimentally and theoretically. In particular, the (potentially positive) effects of (individual) exploration to social welfare (here, fair distribution of rewards) constitute a concrete and intriguing direction for future research in this area.