# OpenReview forum: "Exploration-Exploitation in Multi-Agent Competition: Convergence with Bounded Rationality"
_NeurIPS.cc/2021/Conference — NeurIPS 2021 Spotlight_

### Official Review · Reviewer_xRAX · 2021-07-16

**Rating:** 6
**Confidence:** 3

**Summary:**

This paper studies the convergence of Boltzmann Q-learning to the unique quantal-response equilibrium (QRE),   in weighted zero-sum polymatrix games, assuming all agents using positive exploration rates.

**Ethical Concerns:**

No visible ethical issues.

**Limitations And Societal Impact:**

No visible negative societal impact.

**Main Review:**

Strengths:

This paper provides a solid contribution to the literature on selecting Nash equilibria in multi-agent systems, not only in cooperative settings but also in competitive settings. Considering the complexity of multi-agent systems and the minor assumptions needed in this paper,  the present paper serves as a good start to understanding the effectiveness of exploration in the convergence of Q-learning in multi-agent systems. The experiments on some zero-sum games show the convergence of the algorithm and the necessity of exploration.

Questions:
- In the experiment ''Two-agent Weighted Zero-Sum Games'', what's the performance if both of the agents use ETE? I wonder whether both of the agents doing exploration at the beginning stage would affect the convergence.
- Can the convergence be observed in larger games and would it be robust? Currently, the simulation is in a 7-agent game. What's the potential difficulty of simulating the proposed method in larger games?


**Time Spent Reviewing:**

2.5

---

> ### Author Response · Authors · 2021-08-10
> **Response to Reviewer xRAX's official review**
>
> We thank Reviewer xRAX for their positive evaluation of our paper and their concise comments.
>
> Question 1.
>
> This is indeed an interesting question and we explore precisely this question in the provided supplementary material. The result is visualized in the right panel of Figure 6 in Appendix C.1 (Figure 6 complements Figure 1 of the main part: both figures exhaust all combinations between CLR-1 and ETE). Figure 6 shows that when both agents use ETE, not only do both converge (as guaranteed by our main result, Theorem 4.1), but also that the path on the QRE surface becomes straightforward. In particular, since both agents explore in the beginning of the learning period, they both soon reach a unique QRE close to the uniform distribution which leads to *simpler* learning path over the QRE surface.
>
> Importantly, and going beyond the games that are studied in the experiments, our theoretical result guarantees that convergence will be the case in any game, and in particular convergence is robust over various parameters such as exploration policy and the rate of decrease/increase in exploration rates (polynomial or linear).
>
> Question 2.
>
> We thank Reviewer xRAX for raising this point and we appreciate the opportunity to report our experimental results in larger games but more importantly, to highlight the implications of our theoretical results in this context. Concerning the first part, we remark that the choice of 7 agents in the ZSPG that we present is only due to space constraints and without any loss of generality. The game is large enough to exhibit all interesting complexities (e.g., that if some agents do not explore, then convergence may be lost) but also small enough to allow for easily presentable outputs. Simulations in this game scale to arbitrary numbers of agents (we have experimented with instances with as many as 50 agents and 100 runs which terminate in 1-2 minutes using Matlab 2020b on a standard laptop: thus, even larger simulations are clearly possible). This remark is well-received and we will report more experiments in the updated version (most likely in its supplementary material).
>
> Concerning the second part, this is a great opportunity for us to point out the importance of having theoretical guarantees of (provable) convergence of the algorithm to QRE. In particular, our main result (Theorem 4.1) implies that smooth Q-learning will converge fast to the (unique) QRE regardless of the size of the game (i.e., number of players, actions, magnitude of exploration rates etc.). The only limitations in the implementations of larger games are the usual ones: running time and computer precision. But other than that, there are no other theoretical reasons to observe non-convergence to QRE or slow convergence in games of any size. Having provable convergence is not common in these complex multi-agent settings and thus, establishing theoretical convergence guarantees for an algorithm (here Q-learning) can be useful for both theorists and applied researchers in the field.
>
> ------------------
> Again, we thank reviewer xRAX for their concise comments to improve our paper and we will be happy to incorporate the above remarks in an updated version of our manuscript.

---

> > ### Comment · Reviewer_xRAX · 2021-09-01
> > **Appreciate the authors' responses**
> >
> > I appreciate the authors' efforts and I'm satisfied with the responses to both of the questions.

---

### Official Review · Reviewer_kq4E · 2021-07-16

**Rating:** 8
**Confidence:** 3

**Summary:**

This paper investigates the convergence of a smooth variant of Q-learning dynamics in multiplayer network games called weighted zero-sum polymatrix games. The authors suggest interpreting the exploration rate of Q-learning as the inverse of the lambda parameter of the quantal response model, determining the degree of the agent’s rationality. The main result of this work claims that in weighted zero-sum polymatrix games, the Q-learning dynamics will always converge to the unique quantal response equilibrium associated with strictly positive exploration rates exponentially fast, irrespective of the initial profile of strictly mixed strategies (i.e., no agent adopts a pure strategy). The manuscript is concluded by empirical analysis of convergence of Q-learning in two-player and multiplayer weighted zero-sum games. The results show that with strictly positive exploration rates, the dynamic will indeed converge to the associated quantal response equilibrium and approximate Nash equilibrium as exploration rates approach zero. In the case of the two-player setting, the authors observe the quantal response equilibrium is reached even when only one of the players explore (and the dynamic is cyclic when no one explores), which is turned into a formal proposition in appendix C for two-player games with two actions per each player. For multiplayer games, exploration by only a subset of players will not guarantee to reach equilibrium.

**Ethical Concerns:**

I do not have any ethical concerns.

**Limitations And Societal Impact:**

I can not think of any limitation not addressed by the authors.

**Main Review:**

The results the authors present in this work are indeed interesting; the inverse of the lambda parameter was interpreted previously as the intensity of noise, and it is compelling to see the interpretation based on exploration. In the limited time allocated for reviewing, I went through the proofs presented in the appendix (and partially also in the main text), and as far as I can tell, they seem correct to me. I am convinced the quality of results meets the standards of NeurIPS, and analyzing the convergence of different variants of Q-learning is definitely worth investigating. The manuscript is well written in terms of prose, the relevant concepts are well described, and the notation is clear. I suggest the authors allocate some space in the introduction to motivate the class of games they study with some real-world examples. I also have a few notes and questions relating mainly to the experimental part of the paper.

What I am missing in the experimental part is some comparison to traditional methods for computing quantal response equilibrium in terms of practical scalability in the number of players and their actions. Even seeing how the convergence is affected by the number of players and sizes of their action spaces for the smooth Q-learning with fixed exploration rates alone would be interesting. Moreover, limiting the exploration rates to zero, as described in the manuscript, closely resembles the tracing procedure introduced by McKelvey and Palfrey in their 1995 paper, for which Turocy presents a practical algorithm in his paper “A dynamic homotopy interpretation of the logistic quantal response equilibrium correspondence”. The algorithm is implemented in the Gambit library. I believe that comparing Turocy’s method to smooth Q-learning with ELE or CLR-1 strategy would also provide insight into how smooth Q-learning performs in solving the equilibrium selection problem in practice.

McKelvey and Palfrey call the selected Nash equilibrium when lambda approaches infinity (or equivalently, the exploration rates go to zero) the limiting logit equilibrium. Turocy claims that in 2x2 games, the limiting logit equilibrium is risk-dominant (and a similar result is also achieved by Anderson, Goeree, and Holt in minimum-effort coordination games), but Zhang and Hofbauer later disputed this. Do the authors have some intuition which equilibrium would be selected, e.g., in the ZSPG game with many players, besides what is described in part C.5 of the appendix?

What do the authors consider to be the main takeaway from the experiments? Is it the fact that strictly positive rates are indeed necessary for multiplayer games to guarantee convergence? Is the convergence lost in the ZSPG game even when all but one player explore?

### After rebuttal

I would like to thank the authors for their detailed response to my questions, and for pointing me to the work of Leonardos and Piliouras. The assumption that the limiting equilibrium will exhibit a higher entropy than other Nash equilibria is intriguing yet natural and I hope there will be some results in this direction in the future.


**Time Spent Reviewing:**

25

---

> ### Author Response · Authors · 2021-08-10
> **Response to Reviewer kq4E's official review**
>
> We thank Reviewer kq4E for their strong support of our paper and for their constructive feedback on our paper.
>
> The suggestion to better motivate the class of games (weighted network zero sum games) with some real-world examples is well-received. As Cai and Daskalakis state in their paper, *On Minmax Theorems for Multiplayer Games*, these games *can be generally used to model a broad class of competitive environments where there is a constant amount of wealth (resources) to be split among the players of the game*. More importantly, current ML and AI research does offer some additional and very actively studied examples: these comprise GANs with multiple generators and multiple discriminators (in which payoffs are governed by pairwise interactions), see e.g., MGAN: Multi-Generator Generative Adversarial Nets (https://arxiv.org/abs/1708.02556) and MD-GAN: Multi-Discriminator Generative Adversarial Networks for Distributed Datasets (https://arxiv.org/abs/1811.03850) and generally, actor-critic systems with multiple actors/critics. We agree that including these examples in the Introduction will better motivate this class of games to a more broad range of readers.
>
> Concerning the main takeaways from the experiments, these are indeed (1) to experimentally verify that our theorem is tight (Figures 3 and 5) and (2) to build intuition by visualizing the QRE surface in low dimensional games (Figures 1, 2) and possibly (by projections) also in larger games (Figure 4). As correctly pointed out by the reviewer, the ZSPG example shows that if exploration is performed by only some agents, then this may not be enough to ensure convergence even in non-trivial, connected networks.
>
> We thank the reviewer for bringing to our attention the results of Turocy and Zhang and Hofbauer in 2x2 games. This connection raises a very interesting point and we can report the following. Recent results by Leonardos and Piliouras (see reference [34] in the submitted paper) actually confirm the finding of Zhang and Hofbauer: even in the simplest possible setting of 2x2 coordination games, the outcome of Q-learning is path-dependent. In particular, in such games, the Q-learning dynamics converge to the risk-dominant equilibrium if the QRE surface is disconnected (e.g., in Stag Hunt or Pareto Coordination). If the QRE surface is connected (e.g., Battle of the Sexes), then the outcome of the dynamics is indeed a-priori ambiguous (it may/may not converge to the risk-dominant equilibrium) and depends on the exploration policies of the 2 agents.
>
> In contrast to the above, the problem of equilibrium selection in 2x2 weighted zero-sum games is simple: unlike coordination games, these games have a unique QRE for all positive exploration rates (see e.g., the QRE surfaces in Figure 1), and as exploration rates go to zero, these QREs approach the unique (interior) Nash equilibrium of the unperturbed game. In larger (weighted network zero-sum games) games, the problem of equilibrium selection becomes much more relevant and highly nontrivial. These are complex settings with multiple equilibria and deriving analytic characterizations is highly unlikely (much like the lack of characterizations in cooperative settings beyond the 2x2 case). Studying these games (that are characterized by scarce payoff matrices due to the pairwise interactions) in the graphical interface of Gambit requires hardcoding payoff matrices that increase exponentially in the number of the agents. The communication complexity of our algorithmic approach for such network games is roughly $NM^2$ whereas, in general, a representation that does not explicit use this graph structure is roughly $NM^N$. Together with the theoretical guarantees for fast convergence, this allows the study of larger games. We will make our code (which we submitted as supplementary material) freely available to allow more comparisons in this direction.
>
> In response to this comment, we searched the related literature and came across the paper "Markov Quantal Response Equilibrium and a Homotopy Method for Computing and Selecting Markov Perfect Equilibria of Dynamic Stochastic Games". This paper extends the literature on homotopy methods in a Markov setting. The main result is that every finite dynamic stochastic game has a unique limiting logit Markov QRE which constitutes a Markov perfect equilibrium of the game. Based on this, the authors argue that this homotopy method can be used as a computation method or selection criterion, although this paper does not resolve which equilibrium will be selected.   This does indeed make such a comparison of methods very interesting and an avenue for future work.
>
> This brings us to the reviewers' next question, which is arguably the most important open question from our current research (remains open also in the above referenced paper): can we say something about the Nash equilibrium that will be selected in the limit? This is exactly the point of appendix C.5 (we are particularly thankful to the reviewer for reading up to that point in our supplementary material) and indeed, we don't have a conclusive result yet. However, the experimental results hint towards the following conjecture: as agents increase their exploration rates, their choice distributions converge to QREs with high entropy (high exploration). Subsequently, when they gradually decrease their exploration rates back to 0, the path that starts from this "maximum entropy" point leads to Nash equilibria (at zero exploration) that exhibit a more "fair" split in the rewards between agents (last panels in Figure 10). In a nutshell, one possible conjecture is that entropy plays the role of the tie-breaker and that this process may select in the limit Nash equilibria that exhibit higher entropy. This is indeed a rather interesting open question in this line of research.

---

### Official Review · Reviewer_SnP7 · 2021-07-19

**Rating:** 6
**Confidence:** 2

**Summary:**

This paper studies the convergence of simple bandit learning algorithm in weighted zero-sum games, to the Quantal Response Equilibria. I have several major questions regarding the problem setup, motivation and conclusion.

**Ethics Review Area:**

["I don’t know"]

**Limitations And Societal Impact:**

Not apply

**Main Review:**

This paper studies the convergence of a simple bandit learning algorithm in weighted zero-sum games, to Quantal Response Equilibria. I have several major questions regarding the problem setup and conclusions.

1) Problem setup: The "Q-learning" algorithm defined in Eq (5) is actually a bandit learning algorithm, as it does not consider the transition dynamics (simple repeated game). However, the authors didn't clearly state this is a bandit setup, and mentioned MDP in line 111, which confused the readers a lot.

2) Main result: For the main convergence result part (Section 4), in line 222, the author mentioned the bandit learning will converge to QRE, and will approximate a NE. The first claim is proved, but why this QRE can approximate a NE? QRE and NE are two different solution concepts in games, and one does not imply the other. The authors should theoretically explain the latter statement.

3) Motivation of this work: the author mentioned equilibrium selection is mostly discussed in cooperative game setting, and this is the first work studies these concepts in multi-agent competitive setting. Equilibrium selection in cooperative games make sense, as all agents are collaborative and can coordinate to maximize certain central utility. But it seems equilibrium selection in competitive games doesn't make much sense? Agents are self-interested and the outcome of the game cannot be selected - it's determined by agent's self-interested actions.

4) Experiment: Similar to my question (2), the authors mentioned the learning dynamics converge to Nash equilibrium in all simulated game. But the main convergence result is about the Quantal Response Equilibria  - not the Nash equilibrium. The author should either show results about the convergence to QRE, or clearly explain why convergence to QRE implies convergence to NE in these example games.

**Time Spent Reviewing:**

2

---

> ### Author Response · Authors · 2021-08-10
> **Response to Reviewer SnP7's official review**
>
> We thank Reviewer SnP7 for their efforts in reading our paper. Below are our responses to their comments.
>
> 1. The algorithm that we study is not related to bandit learning. It is a continuous time dynamic with unlimited resources to update its Q-values before updating its choice distributions. This is also explained in the references that we cite and which study the same algorithm:
>
>     Sato and Crutchfield [46,47] (we follow the enumeration in the references of our submission),
>
>     K. Tuyls, K. Verbeeck, and T. Lenaerts [50],
>
>     D. H. Wolpert, M. Harré, E. Olbrich [56] and
>
>     P. Mertikopoulos and W. H. Sandholm [38] among many others.
>
> 2. The connection between QRE and NE is well known and is described in the original reference of McKelvey and Palfrey [36] who define QRE and who elaborate on this connection. In particular, in this paper they prove (Theorem 2) that as the exploration rate goes to zero the set of QREs converges to the set of Nash equilibria.  We explain this in lines 178-179 but, the suggestion to add more pointers to that paper to aid the readers (e.g., in line 222) is well-received.
>
> 3. Equilibrium selection refers to the problem of analytically arguing which equilibrium will be realized when multiple equilibria exist and not in exogenously "selecting" (enforcing or imposing) an equilibrium to the agents (as interpreted by the reviewer). This is a particularly relevant problem in current multi-agent research in  both cooperative and competitive settings (except of zero-sum games in which all equilibria yield the same utility to the agents). We explain this in the Introduction of the paper and provide references to papers that study the same problem. In fact, such tricks derive homotopy methods for computing/selecting Nash equilibria as also nicely pointed out by Reviewer kq4E.
>
> 4. Figures 1,2,4 and 5 all show convergence of the learning dynamics to QRE (positive exploration rates) and not to Nash equilibria (in fact, Figure 1 also the QRE surface for better intuition). More experiments (showing convergence to QRE) are included in the appendix.

---

> > ### Comment · Reviewer_SnP7 · 2021-08-22
> > **Thanks for the authors' response**
> >
> > Thanks for the authors' response. Sorry for my misunderstanding about the problem setting. I am happy to increase my score after the clarification.

---

### Decision · Program_Chairs · 2021-09-27

**Decision:**

Accept (Spotlight)

**Comment:**

This paper shows that for a class of multi-agent competitive games -- weighted zero-sum polymatrix games, a variant of Q-learning can provably converge to the unique quantal response equilibrium.
(+) The paper presents a set of new theoretical results regarding convergence guarantees of learning rules in multi-agent games, an area that is understudied so far despite the popularity of learning algorithms used for multi-agent settings. The theoretical analysis is solid.
(+) The numerical experiments show the learning dynamics in some example games, which confirms the theoretical results.
(-) There are some minor presentation issues, for example, the term “network competition” is used without explanation in the introduction, and the use of “.” instead of “,” in line 67.
(-) The reviewers also made a number of suggestions to further strengthen the paper, including adding additional motivations, connections between QRE and NE, additional experiments regarding the scalability.

Overall, the reviewer team has a positive view of the paper. We suggest the authors make changes they promised in the responses to improve the paper.